

# Investigating the role of typhoon-induced gravity waves and stratospheric hydration in the formation of tropopause cirrus clouds observed during the 2017 Asian monsoon

Amit Kumar Pandit[1], Jean-Paul Vernier[1,2], Thomas Duncan Fairlie[2†], Kristopher M. Bedka[2], Melody A. Avery[2], Harish Gadhavi[3], Madineni Venkat Ratnam[4], Sanjeev Dwivedi[5], Kasimahanthi Amar Jyothi[6], Frank G. Wienhold[7], Holger Vömel[8], Hongyu Liu[1,2], Bo Zhang[1,2], Buduru Suneel Kumar[9], Tra Dinh[10], and Achuthan Jayaraman[11]

[1]National Institute of Aerospace, Hampton, USA
[2]NASA Langley Research Center, Hampton, USA.
[3]Physical Research Laboratory, Ahmedabad, India.
[4]National Atmospheric Research Laboratory, Gadanki, India.
[5]Meteorological Centre, India Meteorological Department, Ministry of Earth Sciences, Bhubaneswar, India.
[6]National Centre for Medium Range Weather Forecasting, Noida, India.
[7]ETH Zurich, Zurich, Switzerland.
[8] National Centre for Atmospheric Research, Boulder, USA.
[9]Tata Institute of Fundamental Research Balloon Facility, Hyderabad, India.
[10]Department of Physics, University of Auckland, New Zealand.
[11]Bangalore University, Bangalore, India.

*† Deceased on 27 April 2022*

*Correspondence to:* Amit Kumar Pandit (amitkpandit86@gmail.com)

**Abstract.** We investigate the formation mechanism of a tropopause cirrus cloud layer observed during the Balloon measurement campaigns of the Asian Tropopause Aerosol Layer (BATAL) over Hyderabad (17.47° N, 78.58° E), India on 23 August 2017. Simultaneous measurements from a backscatter sonde and an optical particle counter onboard a balloon flight revealed the presence of a subvisible cirrus cloud layer (optical thickness ~0.025) at the cold-point tropopause (temperature ~ -86.4 °C, altitude ~17.9 km). Ice crystals in this layer are smaller than 50 μm with a layer-mean ice-crystal number concentration of about 46.79 $L^{-1}$. Simultaneous backscatter and extinction coefficient measurements allowed us to estimate range-resolved lidar ratio inside this layer with a layer-mean value of about 32.18±6.73 sr which is in good agreement with earlier reported values at similar cirrus cloud temperatures. The formation mechanism responsible for this tropopause cirrus is investigated using a combination of three-dimensional back-trajectories, satellite observations, and ERA5 reanalysis data. Satellite observations revealed that the overshooting convection associated with a category-3 typhoon *Hato*, which hit Macau and Hong Kong on 23 August 2017 injected ice into the lower stratosphere. This caused a hydration patch that followed the Asian Summer Monsoon anticyclone to subsequently move towards Hyderabad. The presence of tropopause cirrus cloud layers in the cold temperature anomalies and updrafts along the back-trajectories suggested the role of typhoon-induced gravity waves in their formation. This case study highlights the role of typhoons in influencing the formation of tropopause cirrus clouds through stratospheric hydration and gravity waves.



## 1 Introduction

Cirrus clouds are composed of non-spherical ice crystals that exhibit high variability in their size and shapes. Optically thin cirrus coverage is highest over the tropics in a region between the top of maximum convective outflow (near 12-14 km) and the cold-point tropopause (CPT, between 16 and 18 km) called the Tropical Tropopause Layer (TTL; Randel and Jensen (2013). Due to their frequent occurrence at high altitudes and cold temperatures, and their thin-wispy structures, cirrus clouds trap long-wave terrestrial radiation more efficiently than they reflect incoming short-wave solar radiation (Lohmann and Gasparini, 2017). Therefore, high thin cirrus clouds induce a net warming impact on the climate system (Gasparini and Lohmann, 2016; Hong et al., 2016). The radiative effects of cirrus clouds depend on their macrophysical (coverage, altitude, geometrical thickness) and microphysical properties (number concentration, mass density and size and shape distributions of ice crystals) (Liou, 1986, 2005). Higher cirrus clouds have a larger warming effect on climate than those at lower levels (Lohmann and Gasparini, 2017). Simulations using General Circulation Models (GCMs) have shown that the radiative impact of cirrus clouds is sensitive to small changes in the number concentration of small ice crystals (Sanderson et al., 2008; Mitchell et al., 2008). In this context, subvisible cirrus clouds (having optical thickness less than 0.03 at visible wavelengths) occurring near the CPT are important because they are the highest type of cirrus clouds with significant horizontal coverage. Subvisible cirrus clouds can have a relatively long lifetime because they usually consist of low concentrations of small ice crystals, which are more slowly removed by sedimentation and yield greater radiative heating than larger ones, given the same ice mass (Fu and Liou, 1993).

Apart from their radiative impact, subvisible cirrus clouds play an important role in regulating the water vapour in the upper troposphere and lower stratosphere (UTLS) region through dehydration (Jensen et al., 1996). Dehydration efficiency depends on microphysical processes such as nucleation, growth and sedimentation (Rollins et al., 2016). Understanding of microphysical processes under different dynamical forcings and their representation in GCMs remain poor due to numerous factors, one being the lack of accurate measurements of microphysical properties of ice crystals with particle sizes less than 100 µm from the current generation probes (Heymsfield et al., 2017; Baumgardner et al., 2017; Kuhn and Heymsfield, 2016). In situ measurements provide accurate information about the microphysical properties, which are essential for estimating their radiative impact on climate and for the calibration and validation of ice-cloud products from passive and active remote sensing sensors. Balloons and aircrafts are the most common in situ measurement platforms for ice crystal microphysical properties. Due to their slow ascent, balloons provide cloud vertical profiles at high resolution, unlike aircrafts which provide larger horizontal coverage but miss fine vertical information and detailed imaging of ice crystal habits. Moreover, ice measurements using balloons do not suffer as much from shattering effects and do not require corrections due to compressed air as in case of aircraft measurements under its wings (Wolf et al., 2018).

There have been several in situ measurements of tropical cirrus clouds using balloon-borne and aircraft-based instruments (Krämer et al., 2016; Jensen et al., 2017; Schoeberl et al., 2019; Krämer et al., 2020 and references therein). However, most of those measurements were over Australia, Brazil, Central America, Europe, and the Western Pacific. Only a limited number of field campaigns have occurred over the Asian Summer Monsoon



(ASM) region (Krämer et al., 2020), despite the important role played by ASM in the transport of aerosols (Vernier et al., 2015, 2018), water vapour (Nützel et al., 2019; Khaykin et al., 2022; Wang et al., 2019) and the formation of cirrus clouds (Ueyama et al., 2018) in the UTLS region. Additionally, the ASM region has the largest coverage of high-altitude cirrus and sub-visible cirrus clouds during the boreal summer (Martins et al., 2011). Field campaigns such as the Balloon measurement campaigns of the Asian Tropopause Aerosol Layer (BATAL) (Vernier et al., 2018) and aircraft campaigns such as StratoClim (Krämer et al., 2020) and the Asian Monsoon Chemical and Climate Impact Project (ACCLIP, https://www.eol.ucar.edu/field_projects/acclip) have been organized over the ASM region to understand the physical, chemical, and dynamical characteristics of the Asian Tropopause Aerosol Layer (ATAL) along with cirrus cloud microphysics.

Satellite measurements have shown that the UTLS aerosol optical depth between 13 and 18 km altitude over the ASM region has increased three-fold since the late 1990s (Vernier et al., 2015). A balloon-borne study over Tibetan Plateau (He et al., 2019) also showed evidence for the hygroscopic growth of particles inside the ATAL. These may have significant impact on the microphysics of cirrus clouds, especially subvisible cirrus clouds that occur in the proximity of CPT during the ASM (Vernier et al., 2018). In this context, long-term (1998-2013) lidar observations over a tropical station Gadanki (13.5° N, 79.2° E) in south India have shown an increase in the fraction of subvisible cirrus clouds (Pandit et al., 2015). Also, the StratoClim campaign measurements in July and August 2017 from Nepal have shown the presence of solid ammonium nitrate particles from surface ammonia sources (Höpfner et al., 2019), which act as efficient ice-nuclei in the presence of ammonium sulphate particles as shown in the cloud chamber experiments (Wagner et al., 2020). In general, anthropogenic activities are expected to influence the occurrence and properties of cirrus clouds by meteorological, aerosol-induced or cloud-induced changes (Kärcher, 2017). However, these changes over the ASM region, which is one of the most polluted regions of the globe, are not well known and warrant further investigations.

The UTLS region during the ASM is dominated by the complex interplay among frequent deep convection, gravity waves, and large-scale updrafts which directly and/or indirectly lead to the formation of cirrus clouds either through liquid-origin clouds or through in situ freezing mechanisms (Krämer et al., 2016). These dynamical processes directly influence the microphysical properties of cirrus clouds. For example, slow updraft leads to thinner cirrus clouds with low IWC while faster updraft leads to thicker cirrus clouds with higher IWC (Krämer et al., 2016). A study (Ueyama et al., 2018) done using a one-dimensional (vertical), time-dependent cloud microphysical model, diabatic back-trajectories, and observations of convective clouds has suggested that nearly all the enhancement of water vapour and clouds at 100 hPa level over the ASM are due to convective saturation with minimum impact from the convectively detrained ice. This is in contrast to another recent study (Wang et al., 2019) where convective lofting of ice during the ASM is found to be the most important source of water vapour at 100 hPa level in the 10-40° N region. Khaykin et al. (2022) have recently shown evidence of direct convective hydration (water vapour mixing ratio >10 ppmv) by the overshooting convection over the ASM region using StraoClim aircraft measurements. Such large-scale organized convective systems in the southern ASM anticyclone are also associated with synoptic-scale dehydration near the tropopause. In this context, the role of large-scale organized



deep convection over the East Coast of India in the formation of tropopause cirrus clouds is a subject of interest for the BATAL campaigns organized in Hyderabad (17.47° N, 78.58° E), India (see Sect. 2.1).

In addition to meso-scale deep convection, overshooting convection frequently occurs in the tropical cyclones (Romps and Kuang, 2009), especially during their intensification (Horinouchi et al., 2020). Tropical cyclones (typhoons) during the ASM have also been seen to cause dehydration near the CPT (Li et al., 2020) and hydration in the lower stratosphere (Jiang et al., 2020). Climate change is expected to strengthen and increase the occurrence frequency of such typhoons (Stocker et al., 2013). Extreme tropical convection is expected to impact TTL water vapour and cirrus clouds (Aumann et al., 2018). However, a recent climate model study (Smith et al., 2022) suggests that the role of convective ice injection to the stratospheric water vapour budget will remain constrained by large-scale temperatures in a warmer climate. Thus, a change in large-scale temperature fields in the TTL such as those created by tropical cyclones (Biondi et al., 2015) or intense convective systems (Biondi et al., 2012) will determine how much injected ice will be converted into water vapour in addition to injection altitude and frequency of convective penetration. Thus, it is essential to understand how such systems influence the large-scale temperature fields and hence cirrus cloud formation in the TTL. In this context, the ASM, which is influenced by frequent deep convective systems and tropical cyclones, is preferred region for studying such influence on cirrus clouds as presented here.

This paper presents a case study of in situ measurements of microphysical and optical properties of a tropopause cirrus cloud layer observed during the BATAL campaign using balloon-borne instruments (a backscatter sonde and an optical particle counter) capable of detecting very thin cirrus clouds consisting of ice crystals smaller than 100 µm. We also demonstrate the usefulness of combined measurements from these instruments in measuring the range-resolved lidar ratio in situ for a tropical tropopause cirrus cloud layer for the first time (to the best of our knowledge). Using back-trajectory calculations and their intersection with the convective clouds observed by radiometers on the Himawari-8 geostationary satellite, we also investigate the origin of the air masses sampled during the balloon flight. The formation mechanism and properties of tropopause cirrus clouds along the back-trajectories are examined using the satellite observations. Section 2 describes the BATAL campaign, balloon-borne instruments, satellite, and model data used in this study. Section 3 discusses the results and Sect. 4 contains the summary.

**2 Data and method**

**2.1 BATAL campaign and general meteorological conditions**

The balloon measurements were carried out under the framework of the India Space Research Organization (ISRO)-National Aeronautics and Space Administration (NASA) joint project called BATAL (Vernier et al., 2018) during August 2017 from Tata Institute of Fundamental Research Balloon Facility (TIFR-BF, https://www.tifr.res.in/~bf/) located at 17.47° N, 78.58° E, in Hyderabad, India. During the ASM period (June to

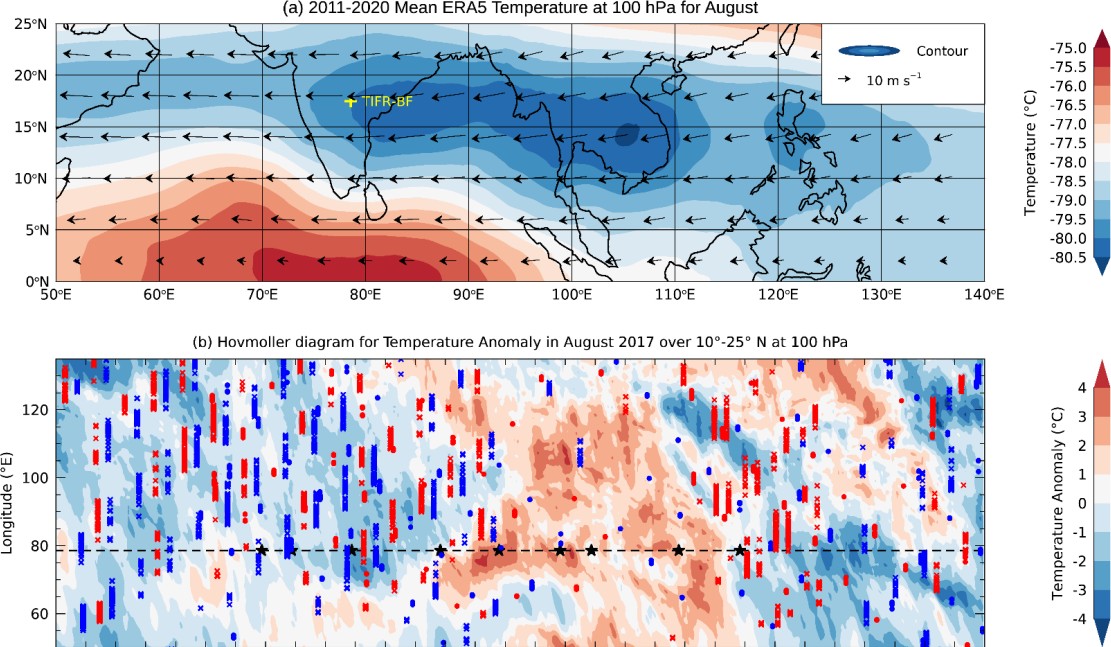

**Figure 1: (a) Filled colour contour showing the 10-year (2011-2020) mean ERA5 temperature superimposed with wind vectors representing the mean ERA5 horizontal wind speed and direction at 100 hPa pressure level for the month of August. The yellow "plus" sign indicates the location of the TIFR-BF in Hyderabad, India. (b) Hovmöller plot for the anomaly in temperature from ERA5 temperature at 100 hPa pressure level for August 2017. The anomaly is computed by subtracting the monthly mean (for August 2017) temperature for each grid between 10° N and 25° N. The dashed horizontal black line marks the longitude (78.58° N) of the TIFR-BF with each black star on it showing the launch time of the balloon. The red (blue) filled circles and crosses superimposed on the filled colour contours represent the locations of cirrus cloud layers detected by CALIOP and CATS during daytime (nighttime), respectively, with their base altitude located at 16 km or above.**

August), deep convective clouds developing over the Bay of Bengal (BoB) and over the land, especially over the Indian East Coast, cause precipitation in the late afternoon and early evening hours. Climatologically, Hyderabad receives highest rainfall during the months of July and August due to frequent deep convection. Launching a heavy plastic balloon flight from TIFR-BF during this time is thus quite challenging due to strong south-westerly surface winds and frequent precipitation. During this period, the Tropical Easterly Jet (TEJ) with maximum wind speeds of about 40 m s$^{-1}$ prevails over Hyderabad in the UTLS region (Vernier et al., 2018). TEJ brings moisture from the Bay of Bengal, Western Pacific, and Southeast Asia (Das et al., 2011) necessary for cirrus cloud formation. Also, the tropopause over Hyderabad is extremely cold (minimum temperature reaching about -86°C) as observed during the BATAL campaigns (Vernier et al., 2018) as compared to north India. The 10-year (2011-2020) mean temperature at 100 hPa pressure level (~16.5 km altitude) from ERA5 (described in Sect. 2.4.3) during the month of August shows a pool of cold-region (with temperature between -78 °C and -80 °C) between 10° N and 20° N from the South China Sea to the Indian East Coast with Hyderabad located right at the edge (Fig. 1a). This cold



region, often influenced by deep convective activities and the TEJ, seems to be a hot spot for laminar cirrus clouds
as evident from the observations of two space-borne lidars (described in Sect. 2.3.1 and 2.3.2) as shown in Fig. 1(b)
for the month of August 2017. The occurrence of these cirrus clouds with their base altitude at or above 16 km is
associated with negative temperature anomaly estimated from the ERA5 temperature by subtracting the monthly
mean temperature from each grid ($0.25° \times 0.25°$) in this cold pool (Fig. 1b). All these factors make TIFR-BF an
ideal location for cirrus cloud studies during the ASM. A total of 9 balloon flights were launched from TIFR-BF
during August 2017 (Vernier et al., 2018) and the timings of these flights are shown in Fig. 1(b). The balloon-borne
instruments used for measuring the cloud properties are described in Sect. 2.2 below.
**2.2  Balloon-borne observations**
On the night (at 20:03 UTC or 01:33 LT) of 23 August 2017, balloon-borne instruments onboard a Heavy Flight
(HF; see Vernier et al. (2018) for description) were launched from TIFR-BF to measure the vertical distribution of
aerosols, clouds and meteorological parameters as described in the sub-sections below. Satellite and model data
have also been used in this study to support the balloon observations, which are described in the following sub-
sections.
**2.2.1 Compact Optical Backscatter AerosoL Detector (COBALD)**
COBALD is a unique lightweight (about 500 g) balloon-borne backscatter sonde developed at the Institute of
Atmospheric and Climate Science, Swiss Federal Institute of Technology (ETH), Zurich. COBALD enables us to
detect optically thin layers of aerosols and clouds at high vertical resolution (Brabec et al., 2012; Cirisan et al.,
2014; Martínez et al., 2021; Vernier et al., 2015, 2016, 2018; Brunamonti et al., 2018). COBALD consists of two
high power (500 mW) Light Emitting Diodes (LEDs) which emit light at 455 nm (blue) and 940 nm (near infrared)
wavelengths. A silicon photodiode placed between the two LEDs collects the light backscattered from the air-borne
particles (molecules, aerosols, and clouds). When connected with an iMet radiosonde, the instrument adds
backscatter and monitor signals at both wavelengths to the radiosonde pressure, temperature, relative humidity,
wind speed and wind direction data at a rate of 1 Hz to the receiving ground station. The concurrent profiles of
pressure, temperature and backscattered counts are used to calculate backscatter ratio (BSR), the ratio of total
(molecules and particulate) backscatter coefficient ($\beta = \beta_p + \beta_m$) to the molecular backscatter coefficient ($\beta_m$) at a
given wavelength. The BSR is calibrated in a region of the atmosphere minimally affected by aerosols and adjusted
with a reference density signal derived from pressure and temperature. The absolute error associated with BSR is
within 5 % while its precision is better than 1 % in the UTLS region (Vernier et al., 2015). The BSR is an optical
analogue for particle mixing ratio. The ratio of ($BSR_{940}$-1) at 940 nm to ($BSR_{455}$-1) at 455 nm, defined as the colour
index (CI), gives qualitative information about particle size (Cirisan et al., 2014; Vernier et al., 2015). On
substituting the molecular backscatter coefficient at 455 nm and 940 nm wavelengths, CI reduces to
$$CI = 18 \times \left( \frac{\beta_{p940}}{\beta_{p455}} \right) = 18 \times CR \qquad (1)$$



where $\beta_{p455}$ and $\beta_{p940}$ are particulate backscatter coefficients at 455 nm and 940 nm, respectively and the ratio $\frac{\beta_{p940}}{\beta_{p455}}$
is called the particulate colour ratio (CR). CR value near unity indicates the presence of cloud layer whereas values
below 0.7 indicates aerosol presence (Vernier et al., 2015; Brunamonti et al., 2018). Ångström's exponent
estimated from the $BSR_{940}$ and $BSR_{455}$ is used to estimate the backscatter ratio at 532 nm ($BSR_{532}$) following
Vernier et al. (2015). After applying the Field of View (FOV) corrections to COBALD backscatter coefficient as
suggested by Brunamonti et al. (2021), an extinction coefficient profile is obtained at 532 nm wavelength by using
the $BSR_{532}$ profile and assuming a constant lidar ratio for cirrus clouds following the CALIOP L2 V4 algorithm
where initial lidar ratio is a sigmoid function of centroid temperature ± 10 sr (Young et al., 2018). By integrating
these extinction coefficients between the base and top altitudes of the cloud layer, optical thickness (τ) of the cloud
layer is estimated. The extinction coefficient derived from the COBALD is used for estimating the IWC of the
tropopause cirrus cloud layer using a temperature-dependent parametrized relation derived from the aircraft
measurements (Heymsfield et al., 2014) as described in the next sub-section (2.2.2).
**2.2.2 SOLAIR Boulder Counter**
The Solair Boulder Counter is a medium-weight (~6 kg) portable, forward scattering-based particle counter from
Lighthouse Worldwide Solutions, USA (https://www.golighthouse.com/en/airborne-particle-counters/boulder-
counter), which uses an extreme life laser diode at 680 nm as a light source. Particles are sampled at a high flow
rate of 28.3 Litres per minute (LPM) and are directed to the laser beam. The light scattered at 45° angles from the
particles is collected and focused by collection optics onto a photo diode which converts it into electric pulses. The
pulse amplitude is a measure of particle size. The number of pulses as a function of pulse height gives particle
counts and size information. Particles are counted in six size channels between 5 µm and 100 µm viz., 5, 10, 25,
40, 50 and 100 µm with 50 % counting efficiency at 5 µm size and 100 % at other size bins. The Boulder Counter's
ability to measure large particles allows us to infer the distribution of larger air-borne particles such as cloud
droplets and ice crystals from a balloon platform. Particle counts in each bin are recorded onboard every 5 seconds
which when divided by the instantaneous sample flow gives particle number concentration for that size bin at that
time. Boulder Counter measurements were synchronized with COBALD measurements. The flow is controlled and
monitored through an external mass-flow controller system. During the flight, the sample flow decreased gradually
with decreasing pressure, but it remained above 10 LPM up to 20 km altitude. The entire system was adapted for
balloon measurements and enclosed in a foam box to maintain the operating range of conditions during the balloon
flight between 10 ℃ and 40 ℃ of temperature. The system was calibrated at the company before the flight and the
performance of this system was also tested in a thermo-vacuum chamber prior to the balloon flight. To the best of
our knowledge, this system is used for the first time for measuring ice particles in ice clouds from a balloon
platform.

35           Only data during the ascent flight are used for estimating the optical and microphysical properties of cirrus

clouds. To estimate the microphysical (particle number concentration, effective diameter, and IWC) and the optical
properties (extinction coefficient and lidar ratio) from the Boulder Counter data, a log-normal size distribution





function was fit to the cumulative particle number concentration within the cloud layer. A single mode lognormal
size distribution function is expressed as

$$\frac{d}{dr}N(r) = N_0 \frac{\exp\left[-\frac{1}{2}\left\{\frac{ln^2\left(\frac{r}{rm}\right)}{ln^2\sigma}\right\}\right]}{rln\sigma\sqrt{2\pi}}$$
(2)

where $N$ is the number concentration, $N_0$ is the total particle concentration in the mode, $r$ is the radius, $r_m$ is the
median radius, and $\sigma$ is the width of the distribution (Thomason and Peter, 2006). A previous study (Kuhn and
Heymsfield, 2016) has shown that a log-normal distribution provides better match with the data in contrast to a
gamma distribution. The first and the third moments of the size distribution are calculated from equation (2) which
are then used to estimate the effective diameter ($D_e$) and IWC, respectively (described below).
The effective diameter of ice crystals is an important microphysical parameter used for the radiative
calculations of ice clouds. It is defined as the ratio of absorption to the extinction cross-section of the ice crystals
(Foot, 1988). Approximating ice crystals as spheres, $D_e$ can be estimated using a log-normal particle size
distribution from equation (3) as shown below:

$$D_e = 2.\frac{\int_0^r r^3 N(r)dr}{\int_0^r r^2 N(r)dr}$$
(3)

where the numerator and the denominator are the third and second moments of the particle size distribution function,
respectively. We also estimated the effective diameter using its temperature dependence relation given by
Heymsfield et al. (2014) as shown below in equation (4) to compare with that derived from the size distribution.

$$D_e = \alpha e^{\beta T}$$
(4)

where $T$ is the temperature,
$\alpha = 308.4$, $\beta = 0.0152$ for -56 °C $< T <$ 0 °C.
$\alpha = 9.1744 \times 10^4$, $\beta = 0.177$ for -71 °C $< T <$ -56°C.
$\alpha = 83.3$, $\beta = 0.0184$ for -85 °C $< T <$ -71°C.
Using the Airborne Tropical TRopopause EXperiment (ATTREX) campaign data, Thornberry et al. (2017) have
found that the effective diameter decreases more sharply at temperatures (T) below -81 °C (192 K) than previously
found by Heymsfield et al. (2014). This decrease in median effective diameter for T<192 K is expressed as

$$D_e = 12 + 28e^{0.625(T-192)}$$
(5)

We also estimated the extinction coefficient from our particle size distribution measurements, using a geometric
optical approximation which assumes that the extinction coefficient is two times the total cross-sectional area of the
ice crystals. Assuming that the ice particles are close to being spherical, we use the equation for extinction
expressed in Thornberry et al. (2017):

$$\sigma_{ext} = 2 \times \sum_j N_j \pi r_j^2$$
(6)



where $N_j$ is the ice particle number concentration (in L$^{-1}$) corresponding to the particle radius $r_j$ (in µm) with $j$
ranging from 1 µm to 25 µm. The ratio of $\sigma_{ext}$ (from equation 6) to the particulate backscatter coefficient ($\beta_p$)
obtained from COBALD gives lidar ratio (discussed in Sect. 3.3.4).
IWC is an important microphysical parameter for estimating the radiative impact of cirrus clouds as well as for
understanding the microphysical and dynamical processes occurring within them (Heymsfield et al., 2017). We
estimated IWC for the tropopause cirrus independently from the COBALD backscatter and the Boulder Counter
measurements. IWC was obtained from COBALD data using the extinction coefficient ($\sigma_{ext}$) and the effective
diameter ($D_e$) estimated from the Equation 9 (e) given in Heymsfield et al. (2014) as shown below.
$$IWC = \sigma_{ext}\left(\frac{0.91}{3}\right)D_e \tag{7}$$
Here, $D_e$ is obtained from equation (4) using temperature measurements. Using the ATTREX in situ measurements,
Thornberry et al. (2017) have given a new parameterization relation for the estimation of IWC from extinction
coefficient in the temperature (T) range between 185 K (-88 °C)  and 192 K (-81° C) as shown below.
$$IWC = \frac{0.92}{3}\sigma_{ext}\left(12 + 28e^{0.65(T-192)}\right) \tag{8}$$
Total volume obtained from the third moment of the particle size distribution when multiplied with the ice-density
(917 kg m$^{-3}$), it gives IWC. IWC for the tropopause cirrus cloud layer derived from these abovementioned methods
are compared in Sect. 3.3.5.
**2.2.3 India Meteorological Department (IMD) radiosonde profiles from the University of Wyoming**
We use daily (00 UTC) radiosonde profiles of meteorological parameters over the Meteorological centre,
Hyderabad airport (17.45° N, 78.46° E) and its surrounding IMD stations such as Machilipatnam (16.2° N, 81.15°
E), Visakhapatnam (17.7° N, 83.3° E), and Jagdalpur (19.08° N, 82.02° E), obtained from the University of
Wyoming Atmospheric Soundings website (http://weather.uwyo.edu/upperair/sounding.html). Only profiles during
the month of August 2017 are used in this study from these stations. The monthly mean temperature and wind
profiles are constructed by using 27 days of available data over Hyderabad during August 2017.
**2.3 Satellite observations**
**2.3.1 Cloud Aerosol LIdar with Orthogonal Polarization (CALIOP)**
CALIOP was a dual wavelength (532 and 1064 nm), dual-polarization, three channel space-borne lidar onboard the
Cloud-Aerosol Lidar and Infrared Pathfinder Satellite Observations (CALIPSO) orbiting around the Earth in a sun-
synchronous polar orbit at an altitude of about 705 km and an inclination of about 98.2° (Winker et al., 2009). It
has provided optical properties of aerosols and clouds distributed vertically in the Earth's atmosphere for 17 years
since June 2006 at unprecedented spatial resolution with a repeat cycle of 16 days at a given location. CALIOP can
detect thin and subvisible cirrus clouds (Martins et al., 2011) by measuring the attenuated backscatter coefficients
(at two wavelengths and two polarizations), followed by the retrieval of 532 nm extinction coefficients (Young et



al., 2018). IWC and effective diameter are estimated from these retrieved extinction coefficients, using a temperature-dependent empirical fit derived from aircraft data (Heymsfield et al., 2014). In this study, we use CALIOP Level 2 version 4.2 cloud profile (CPro) and cloud layer (CLay) data products having a horizontal resolution of 5 km along the orbit track and 60 m vertical resolution between 8.2 and 20 km altitude range. We obtained scattering ratio profiles from the total backscattering coefficient profiles at 532 nm wavelength and molecular density profiles provided in the CPro data using method described in CALIOP Algorithm Theoretical Basis Document (ATBD, Young et al., 2008).

### 2.3.2 Cloud-Aerosol Transport System (CATS) lidar

CATS was a three wavelength (355 nm, 532 nm and 1064 nm), dual polarization space-borne lidar intended to provide vertical distribution of aerosols and clouds from the International Space Station (ISS) platform (Yorks et al., 2014). It was launched in January 2015, operated until October 2017, and provided 33 months of aerosol and cloud data. The orbit of ISS has an inclination of 51° and an altitude of about 405 km. This geometry provided more coverage over the tropics and mid-latitudes when compared to CALIPSO with a shorter repeat cycle of 3 days and observations at various local times. In this study, we use 5 km horizontally averaged M7.2 Level 2, Version 3.0 Operational Layer (OL) and Profile (OP) products. These products are obtained from one of the two operational modes (M7.2) of CATS in which backscatter coefficient and depolarization ratio at 1064 nm are measured. The horizontal and vertical resolutions are same as CALIOP data. We obtained scattering ratio from the backscattering coefficient measurements at 1064 nm wavelength and molecular density profiles provided in the data using method provided in CATS ATBD (Yorks et al., 2015).

### 2.3.3 Cloud top height from Himawari-8 brightness temperature

Himawari-8 is a geostationary satellite launched by Japan Meteorological Agency in October 2014 to observe weather phenomena in 16 spectral bands (visible, near-infrared and infrared) at high resolution (Bessho et al., 2016). Cloud-top temperature observations from 10.4 μm channel of the Himawari-8 satellite at an interval of 10 minutes have been used as proxy for convection. Low cloud-top temperature indicates the presence of deep convective clouds and anvils. The horizontal resolution of cloud-top temperature is ~ 2 km at satellite nadir for the infrared bands. The influence of deep convection on the tropopause cirrus clouds sampled by our balloon is investigated by using a technique combining back-trajectory analysis (described in Sect. 2.4.1) and cloud-top brightness temperature observations at 10.4 μm from the Himawari-8 developed by Bedka and Khlopenkov (2016). This technique has been used earlier in Vernier et al., (2018) for studying the impact of deep convection including overshooting convection on the air parcels sampled by the balloons.

### 2.3.4 Humidity measurements from Aura-Microwave Limb Sounder (MLS)

Level 2 Version 4.2 relative humidity with respect to ice (RHi) and water vapour mixing ratio (WVMR) data products (Lambert et al., 2015) from the Microwave Limb Sounder (MLS) instrument onboard NASA's Aura





satellite are used. These data products are available at 55 pressure levels between 1000 and $10^{-5}$ hPa. The vertical
resolution of MLS-RHi range from 3.7 to 4.6 km between 68 and 100 hPa with an accuracy from 20 to 25 %. The
vertical resolution of MLS $H_2O$ product is ~3 km with an accuracy of 8-9 % between 68 and 100 hPa levels.
Horizontal resolution along the orbit track is between 190 and 198 km over these pressure levels. The data
screening criteria specified for RHi and $H_2O$ data by Livesey et al. (2017) have been applied to filter out the effect
of clouds.
**2.3.5 Temperature profiles from Global Navigation Satellite System Radio Occultation (GNSS-RO)**
High-resolution Global Navigation Satellite System Radio Occultation (GNSS-RO) dry temperature profile product
(Level 2, atmPrf) obtained from COSMIC-1, Metop A/B, GRACE, KOMPSAT-5, TSX, and TDX missions are
used to find the temperature near the tropopause cirrus clouds detected by CALIOP and CATS. These data are
provided by COSMIC Data Analysis and Archive Center (CDAAC, https://cdaac-
www.cosmic.ucar.edu/cdaac/products.html) at 100 m altitude grids from 0 to 40 km altitude. Comparison of
GNSS-RO temperature profiles with those from radiosonde has shown good agreement (Anthes et al., 2008).
GNSS-RO data set have found many applications in the atmospheric science research such as in the study of the
gravity waves (Liou et al., 2003), thermal structure of the tropical cyclones (Biondi et al., 2013, 2015; Ravindra
Babu et al., 2015), volcanic cloud detection (Biondi et al., 2017), stratospheric thermal perturbation after
2019/2020 Australian pyroCb event (Khaykin et al., 2020) and tropopause cold-anomalies during the 2015 El Nino
event over the Pacific ocean (Ravindrababu et al., 2019).
**2.4 Model simulations and reanalysis data**
**2.4.1 NASA Langley Trajectory Model (LaTM) simulations**
The influence of deep convection on the tropopause cirrus cloud is investigated by using a combination of back-
trajectory analysis and cloud-top temperature observations from the Himawari-8 geostationary satellite (described
in Sect. 2.3.3). Meteorological fields from the NASA Global Modeling and Assimilation Office (GMAO) Goddard
Earth Observing System, Version 5, Forward Processing (GEOS-5 FP; Lucchesi et al., 2017) reanalysis are used to
compute three-dimensional back-trajectories using the NASA Langley Trajectory Model (Fairlie et al., 2009,
2014). Back-trajectories from the location of the balloon measurement are run at every 100 m vertical resolution.
Temperature and wind speeds along the back-trajectories have also been obtained and used in this study to
understand the cloud-formation in the air-masses.
**2.4.2 NOAA HYSPLIT trajectories**
We used National Oceanic and Atmospheric Administration (NOAA) Hybrid Single-Particle Lagrangian Integrated
Trajectory (HYSPLIT) model (https://www.ready.noaa.gov/HYSPLIT.php) to compute forward trajectories of the
air parcels from the cirrus cloud altitudes measured from our balloon payload.



### 2.4.3 ERA5 temperature, relative humidity, and cloud fraction

We use hourly gridded ($0.25° \times 0.25°$) temperature, relative humidity, wind speed and cloud fraction from the ERA5 data product available at 37 pressure levels to understand the spatio-temporal variation of the meteorological variables and to supplement the satellite observations. ERA5 is the fifth generation of ECMWF reanalysis data product, which provides improved representations of the troposphere and tropical cyclones at high spatial and temporal resolution than its predecessor ERA-interim (Hersbach et al., 2020). With reference to the high-resolution GNNS-RO temperature profiles, ERA5 temperature data provide most realistic tropopause temperatures compared to other reanalyses (Tegtmeier et al., 2020). We derive the WVMR using the ERA5 temperature and relative humidity data. The ice saturation vapour pressure is obtained from the relation given by Murphy and Koop (2005). Over the ASM region, the WVMR obtained from ERA5 data provide a good representation of its vertical distribution and variation between 60 and 100 hPa pressure levels with an average overestimation by 0.7-0.9 ppmv (15-17 %) compared to that observed by Cryogenic Frost-point Hygrometer (CFH) during the ASM (Brunamonti et al., 2019).

### 3 Results and discussion
### 3.1 Radiosonde, COBALD, and Boulder Counter observations

This section describes the measurements of clouds and background meteorological parameters from the balloon-borne instruments on 23 August 2017. The trajectory of this balloon flight and the temporal variation of the atmospheric conditions measured by the radiosonde are shown respectively in Fig. S1(a) and (b) in the Supplementary Information (SI hereafter). The vertical profiles of $BSR_{940}$, $BSR_{455}$, and CR obtained from COBALD during the ascent flight show several peaks (Fig. 2(a)). The peaks in the $BSR_{940}$ profile and the corresponding CR values close to unity clearly indicate the presence of five distinct cloud layers labelled as CL1, CL2, CL3, CL4 and CL5, respectively, in the free troposphere from bottom to top (Fig. 2a). Simultaneous measurements from the Boulder Counter shown in Fig. 2(c) reveal the prominence of particles larger than 5 μm (diameter) in these cloud layers. These particles could be water droplets and/or ice crystals depending on the air temperature which is shown in Fig. 2(b) along with the wind speed. In what follows we describe the characteristics of each of these cloud layers from the lowest to the highest.

Specifically, CL1 is a thin mid-level cloud layer (Bourgeois et al., 2016) with its base located at about 4.1 km near the 0 ℃ isotherm and having a geometrical thickness of about 0.7 km. It is also associated with a shallow temperature inversion present near its centre, which might be responsible for its formation as mentioned by Bourgeois et al. (2016). Since CL1 is located at temperature near the 0 ℃ isotherm we do not expect ice crystals in it. This layer is characterized by a low concentration (less than 10 $L^{-1}$) of liquid droplets having size smaller than 40 μm with the majority being between 5 and 10 μm in size. CL2 is a geometrically and optically thick (optical thickness at 532 nm ~ 0.37) cloud layer with cloud base and top altitudes at 8.8 km and 11.3 km, respectively. It consists of a high concentration of particles spread over a wide size range. Since this cloud layer lies between the temperature of -27 ℃ and -45 ℃, it could be a mixed-phase cloud layer having both super-cooled water droplets as



1  well as ice crystals (Korolev et al., 2017). However, its cloud top is colder than -40 ℃ indicating the presence of ice

2  crystals which are smaller than 40 μm. These ice crystals have likely grown larger at the expense

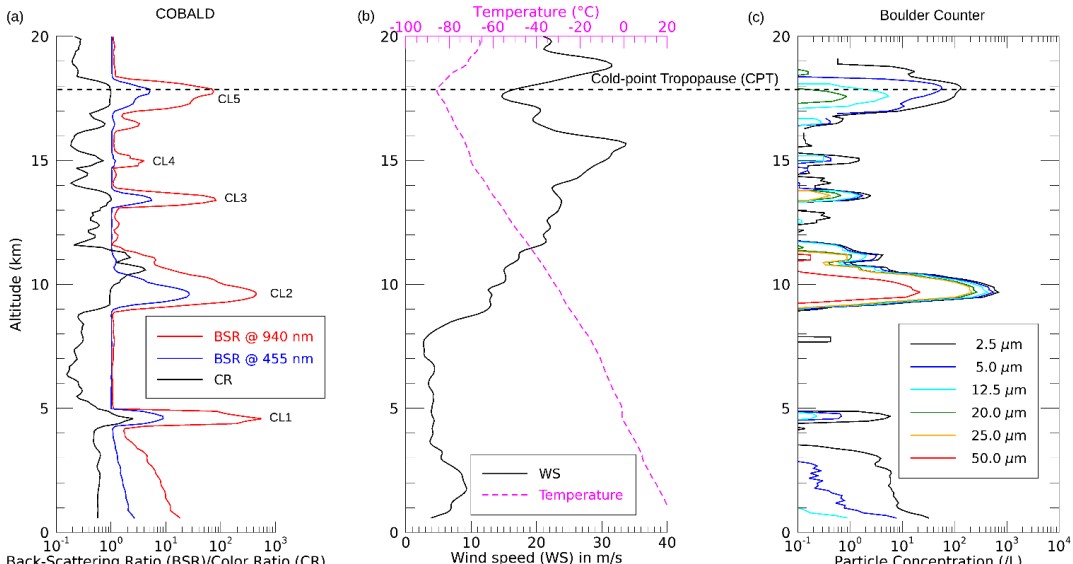

**Figure 2: Vertical profile of (a) backscatter ratio at 455 nm (blue line), 940 nm (red line) and colour ratio (black line) obtained from COBALD, (b) temperature (magenta dashed line) and wind speed (black line) from radiosonde with horizontal dashed black line showing the CPT and (c) cumulative particle number concentration for particles having radius greater than 2.5 (black line), 5 (blue line), 12.5 (cyan line), 20 (green line), 25 (yellow line) and 50 μm (red line) obtained from the Boulder Counter measurements on 23 August 2017 over Hyderabad, India during BATAL campaign. The vertical resolution is 100 m for each profile.**

of ambient supersaturation and consequently have undergone sedimentation, as evidenced from the increased
concentration of larger sized (>40, 50 and 100 μm) ice crystals a few hundred meters below the cloud top and near
the cloud base (Fig. 2c). The peak in $BSR_{940}$ almost coincides with the peaks of particle concentration corresponding
to each size bins. The peak particle concentration (for diameter > 5 μm) in CL2 is found to be about 732.5 $L^{-1}$ at an
altitude of ~9.6 km with the majority of particles sized between 50 and 100 μm and a small number of particles
larger than 100 μm. Above this layer, three layers (CL3, CL4 and CL5) of cirrus clouds (as the temperature is below
-40 ℃) are observed having low concentrations (less than 10 $L^{-1}$ in CL3 and CL4 and less than 200 $L^{-1}$ in CL5) of
ice crystals. In CL3, ice crystals are smaller than 100  μm and in CL4 they are smaller than 40 μm. CL5 is a
subvisible (optical thickness at 532 nm of ~ 0.025) cirrus cloud layer located at the CPT. The CPT is located at an
altitude of about 17.9 km having an extremely cold temperature of about -86.4 ℃ (Fig. 2b). CL5 has a geometrical
thickness of about 2.1 km with its top located at ~18.3 km in the lower stratosphere, ~ 400 m above the CPT. Note
that the peak in $BSR_{940}$ and peaks in particle concentration corresponding to the 5 and 10 μm size bins are located
exactly at the CPT. CL5 consists of particles smaller than 50 μm with most of the particles smaller than 25 μm. This
tropopause cirrus is again detected by the COBALD during the descent flight between 21:49 UTC and 21:51 UTC at
almost similar altitude and similar temperature nearly 100 km southwest from the location of CL5 as shown in Fig.



3(a). About five hours later, a similar tropopause cirrus cloud layer was noticed in the CATS lidar observations as
discussed in the next section. In the subsequent sections we discuss the properties of CL5.
**3.2 CATS lidar observations of the tropopause cirrus cloud layer**

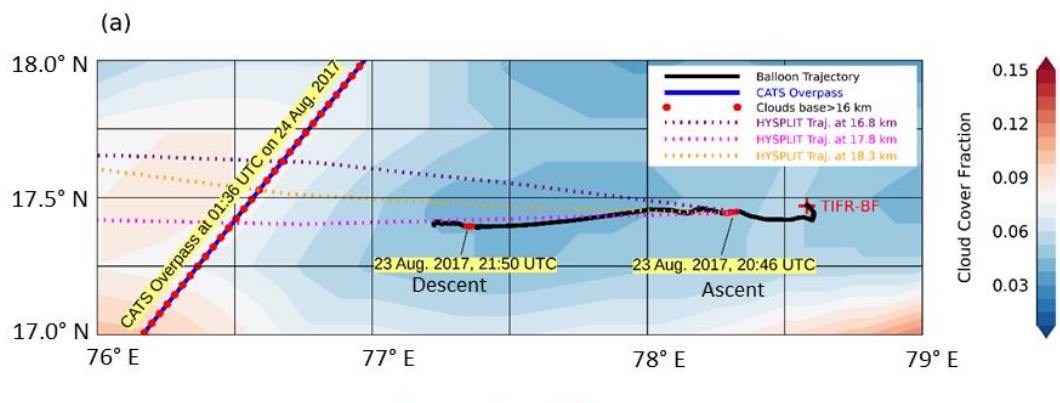

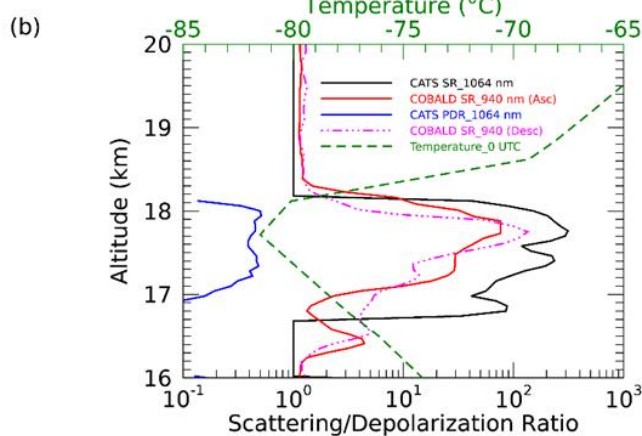

**Figure 3: (a) ERA5 cloud cover fraction at 100 hPa on 24 August 2017 averaged between 01:00 and 02:00**
**UTC. CATS overpass is shown by solid blue line with filled red circles superimposed on it representing the**
**locations of cirrus clouds with base height greater than 16 km. Solid black line shows the balloon flight**
**trajectory from TIFR-BF on 23 August 2017 (shown by red plus sign) with red circles representing the**
**locations of cirrus clouds with base altitude above 16 km during the ascent (at around 20:46 UTC) and**
**descent (at around 21:50 UTC). Dashed lines show the forward trajectories from the location of tropopause**
**cirrus at 16.8 km (purple), 17.8 km (magenta) and 18.3 km (orange), respectively. (b) Vertical profiles of**
**scattering ratio obtained from CATS (at 1020 nm wavelength) and COBALD (at 940 nm wavelength)**
**measurements. CATS profiles are averaged between 17.40° N and 17.65° N region along its orbit track.**
On 24 August 2017 at about 01:36 UTC (~5 hours after CL5 detection), there was an overpass of the ISS nearly 100
km west from the location of the tropopause cirrus cloud detected during the balloon descent flight and ~200 km
from TIFR-BF. This allowed us to corroborate our balloon observations using the onboard operational CATS lidar



data. Linking the data to its larger regional context, Fig. 3(a) shows a colour contour map of mean cloud cover fraction from ERA5 reanalysis data at 100 hPa pressure level (altitude ~16.8 km) pressure level between 01:00 UTC and 02:00 UTC on 24 August 2017. Both balloon (ascent and descent) and CATS measurements of clouds with base altitudes above 16 km at different times and different locations along the wind direction confirm the large horizontal extent of the tropopause cirrus cloud. The horizontal extent of this cloud layer along the CATS orbit track is found to be more than 500 km (Fig. S2).

We further run forward trajectories using HYSPLIT model initialized from different altitudes of the CL5 to see whether the cirrus cloud layer detected by CATS is influenced by the same air mass or not. These trajectories intersect the CATS orbit track between 17.40 ºN and 17.65 ºN. The mean vertical profiles of the backscattering ratio and total depolarization ratio at 1064 nm from the CATS lidar averaged between 17.40 ºN and 17.65 ºN along its orbit track are shown in Fig. 3(b) along with the backscattering ratio profiles from COBALD at 940 nm during the ascent and descent. Although these measurements are at different time, locations and wavelengths, a good qualitative agreement can be seen between the two measurements as far as cirrus clouds near the tropopause are concerned. The value of total depolarization ratio at 1064 nm from the CATS lidar lying between 0.4 and 0.5 confirms that this cloud layer consists of non- spherical ice crystals. The CATS cloud phase algorithm also classifies this layer as an ice-cloud layer (Fig. S2). Daily radiosonde observation at 00 UTC from IMD Hyderabad station (17.45 °N, 78.46 °E) just before the ISS overpass on 24 August 2017 revealed that the CPT altitude is about17.7 km (Fig. 3b) while the CPT temperature is found to be coldest (about -81.5 ºC) in the last one week. This tropopause cirrus cloud layer is the focus of this manuscript, and we discuss about its microphysical (ice crystal number concentration, effective diameter, and ice water content) and optical properties (extinction coefficient and lidar ratio) derived from our balloon measurements in the following section (Sect. 3.3). We also investigate and discuss the mechanisms responsible for its formation in Sect. 3.4.

**3.3 Microphysical and optical properties of the tropopause cirrus cloud**

We got only four altitude bins within the CL5 at 100 m resolution where the estimation of microphysical properties using the size distribution is possible. This is because ice crystals were counted only in a few size bins away from the CPT as shown in Fig. 2(c) and particle concentration in at least four size bins are needed for the solution to converge. Assuming spherical ice crystals in CL5, the log-normal size distributions for particle number and volume concentrations at each of these altitude bins are obtained. Figure 4 shows such distributions for the layer mean particle concentration in CL5. From the size distribution, we derive various parameters as discussed below.



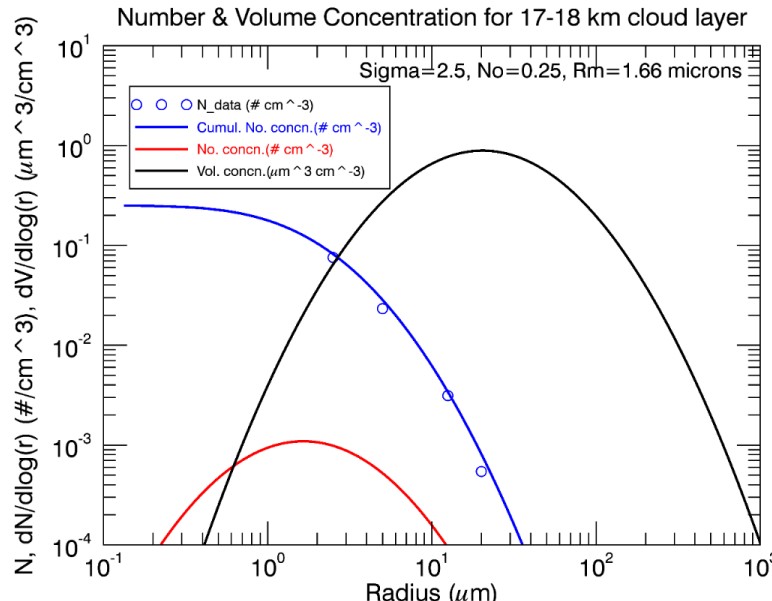

**Figure 4: Particle size distribution of the cloud layer (CL5) observed between 17 and 18 km from the Boulder**
**Counter measurements on 23 August 2017. Particle number concentration and volume concentration are**
**shown by solid red and black lines, respectively. Measured cumulative number concentration is shown by**
**open blue circles while the solid blue line is log-normal fit to it. The width of the distribution is given by**
**Sigma, which is about 2.5, total particle concentration (N$_0$) in the mode is 0.25 per cm$^{-3}$ and R$_m$ = 1.66 µm is**
**the median radius.**
**3.3.1 Ice crystal number concentration (N$_{ice}$)**
As shown in Fig. 4, the cumulative ice crystal number concentration for CL5 derived from the log-normal
distribution (blue curve) fits well with the measured cumulative number concentration (blue circles). The mean
particle number concentration (for diameter > 5 µm) inside the CL5 layer is about 46.79 L$^{-1}$ with a maximum
concentration of ~196.3 L$^{-1}$ at the CPT (Fig. 2c). Particle concentration at different altitude bins within CL5 are
given in Table 1. These values are in good agreement with those measured inside tropopause cirrus clouds under
similar temperatures observed over Bandung, Indonesia (Shibata et al., 2012), the Western Pacific (Woods et al.,
2018) and Costa Rica (Lawson et al., 2008).
**3.3.2 Effective diameter (D$_e$)**
Using equation (3), the effective diameter (*D$_e$*) of ice crystals within CL5 at different altitude bins from 17.3 km to
17.7 km are estimated and are shown in Table 1. We have integrated between the radius from 1 to 25 µm because
we do not see ice particles larger than 50 µm (diameter) in CL5. *D$_e$* varies between 16.34 and 26.61 µm with a
mean plus/minus a standard deviation of 20.22±4.54 µm. The mean effective diameter for CL5 between 17.3 and



17.7 km obtained from temperature using equation (4) is about 18.01±0.49 µm while the value obtained using
equation (5) (i.e., for temperatures between -81.32 ℃ and -85.76 ℃) is found to be about 21.29 µm which is in
good agreement with that obtained from the size distribution. The effective diameter for the tropopause cirrus cloud
layer observed between 17 and 18 km at temperatures below -83 ℃ by Shibata et al. (2007) was in the range from
8 to 80 µm. Whereas the ATTREX data showed that the median value of $D_e$ reaches near 12 µm at temperatures
below -86 ℃ (Thornberry et al., 2017).
**3.3.3 Extinction coefficient ($\sigma_{ext}$)**
The values of the extinction coefficients estimated using relation (6) at different altitude bins within CL5 are given
in Table 1. The mean and standard deviation of $\sigma_{ext}$ (0.028±0.009 km$^{-1}$) are slightly higher than the value
0.018±0.005 km$^{-1}$ obtained from the FOV corrected COBALD backscatter coefficient derived at 532 nm wavelength
using a constant lidar ratio of 22 sr. This value of lidar ratio is chosen based on the CALIOP L2 V4 algorithm
following Young et al. (2018), where the initial lidar ratio at a cloud centroid temperature (for CL5) of about -81.4
℃ is about 22 sr. The lower value of the extinction coefficient derived from COBALD could be due to the choice of
constant lidar ratio.
**3.3.4 Lidar ratio**
The ratio of extinction to backscatter coefficient, often called the lidar ratio, is used in retrieving extinction
coefficient from the backscatter measurements of elastic backscatter lidars. The retrieved extinction coefficient and
hence the layer optical thickness are very sensitive to the magnitude of the lidar ratio used in the extinction retrieval,
which depends on the number concentration, size and shape distributions and refractive index of the particles
(Ackermann, 1998). Since microphysical properties of cirrus clouds mainly depend on temperature and RHi
(Heymsfield et al., 2017), it is essential to study the dependence of lidar ratio on these factors.  The variation of lidar
ratio within a cirrus cloud layer can be measured in situ or by using inelastic lidars, such as Raman lidars (Ansmann
et al., 1992; Sakai et al., 2003) or high spectral resolution lidars (Grund and Eloranta, 1990). In case of elastic
backscatter lidars, for an extinction coefficient retrieval the lidar ratio is usually assumed to be constant (range-
independent) inside a cirrus cloud layer.  Version 3 CATS lidar clouds products use a constant lidar ratio of 28 sr
over land and 32 sr over the oceans for tropical ice clouds (CATS Data Release Notes Version 3.0, 2018). This
approach of using a constant lidar ratio is unable to account for the natural variability of the actual lidar ratio leading
to erroneous extinction coefficient and cloud optical thickness retrievals for cirrus clouds (Saito et al., 2017). Using
multi-year nighttime CALIOP two-way transmission and collocated IIR absorption optical depth at 12.05 µm for
single-layer semi-transparent cirrus cloud layers over the ocean, Garnier et al. (2015) derived temperature dependent
multiple-scattering factor and a corresponding temperature-dependent parametrization of initial lidar ratio.
Following this study, the multiple scattering factor and lidar ratio in CALIOP V4 algorithm are approximated by a
sigmoid function of the centroid temperature of the 532 nm attenuated coefficient within the cloud layer (Young et
al., 2018). According to this sigmoid function, the value of initial lidar ratio for a cirrus cloud layer with centroid



temperature between -80 °C and -90 °C falls in the range of 20-22 sr. The introduction of this approach in the
CALIOP V4 algorithm has resulted in the reduction of the relative uncertainty assigned to the initial lidar ratio for
semi-transparent ice clouds by 15 % compared to V3 algorithm (Young et al., 2018).

4       There are many studies on lidar ratio of cirrus clouds using lidars measurements (see Table 5 of Voudouri

et al. (2020)) but such studies using in situ measurements of subvisible cirrus clouds near the tropical tropopause are
rare. Our unique balloon measurements provided us independent in situ measurements of backscatter (from
COBALD) and extinction coefficients (from Boulder Counter) inside a subvisible cirrus layer near the CPT which
allowed us to estimate the vertical profile of lidar ratio. Table 1 shows the value of lidar ratio inside CL5 at different
altitude bins between 17.3 and 17.7 km with a mean and standard deviation of about 32.18±6.73 sr. In this study, the
mean lidar ratio is estimated for a mean temperature of ~-83.2 °C (see Table 1). To the best of our knowledge, in
situ lidar ratio for cirrus cloud at this low temperature has not been estimated before. Our estimates of lidar ratio are
in good agreement with those values (40±10 sr) obtained by He et al. (2013) for cirrus clouds observed near the
tropopause (temperature ~ -80°C ) over the Tibetan Plateau during July and August of 2011. From this study, we
demonstrate that the combined measurements from COBALD and Boulder Counter can be used to estimate range-
resolved lidar ratio for optically thin cirrus cloud layer near the CPT.

**3.3.5 Ice water content**

The IWC from the size distribution information provided by the Boulder Counter data is estimated by multiplying
the total ice volume by ice density (917 kg m$^{-3}$). By assuming the spherical shape of the ice crystals, the total volume
is estimated from their volume distribution for ice crystal radius ranging between 1 and 25 μm. The values of IWC
estimated at different altitude bins between 17.3 and 17.7 km using different methods are shown in Table 1.  The
mean and standard deviations of IWC estimated from the size distribution between 17.3 and 17.7 km is found to be
about 0.176±0.089 mg/m$^3$. We also estimated the IWC from the FOV corrected extinction coefficient obtained from
COBALD data by using equations (7) and (8) as shown in Table 1. Using equations (7) and (8), the mean and
standard deviations of IWC obtained from COBALD data are respectively found to be ~ 0.100 mg m$^{-3}$ and 0.112 mg
m$^{-3}$, which are in good agreement with that obtained from the size distribution data. When considering the entire
CL5 layer, mean IWC obtained from COBALD data is found to be about 0.055 mg m$^{-3}$ and 0.062 mg m$^{-3}$ using the
relation (7) and (8), respectively. These values are within the range (0.001-10 mg m$^{-3}$) of the IWC for cirrus clouds
observed under similar temperature conditions (-90 °C and -80 °C) during several field campaigns over different
regions of the globe (Lawson et al., 2008; Krämer et al., 2016; Heymsfield et al., 2014; Jensen et al., 2017; Woods et
al., 2018; Thornberry et al., 2017). The IWC (based on the parametrization relation 7) obtained from the CATS lidar
observations between 17 and 18 km altitude, a few hours later of our measurements, is in the range of 0.1-0.6 mg m$^{-3}$
. COBALD and CATS measurements are of the same order, but it is noted that these measurements are at different
wavelengths, different sensitivity levels, different times, and locations. Overall, our measurements are within the
range of IWC measured for cirrus clouds in similar temperature range over other regions using aircrafts and space-
borne lidar. In the next section, we investigate the formation mechanisms of CL5.





**Table 1: Optical and microphysical properties of CL5 derived using COBALD and Boulder Counter**
**measurements on 23 August 2017 from TIFR-BF. Data in *italic* fonts show the properties derived using only**
**Boulder Counter data while data shown in normal fonts are derived from COBALD and/or iMet-1**
**radiosonde data.**

| Altitude (km) | 17.3 | 17.4 | 17.5 | 17.6 | 17.7 | Mean ± Std. dev. |
|---|---|---|---|---|---|---|
| Temperature (°C) | -81.32 | -82.26 | -83.23 | -84.10 | -84.94 | -83.17±1.43 |
| *No. concentration (#/L)* | *54.23* | *67.27* | *99.47* | *101.77* | *108.87* | *85.32±25.12* |
| *$D_{eff}$ (µm) using Eqn. 3* | *16.34* | *-* | *17.81* | *20.11* | *26.61* | *20.22±4.54* |
| $D_{eff}$ (µm) using Eqn. 4 | 18.62 | 18.36 | 18.0 | 17.68 | 17.41 | 18.01±0.49 |
| $D_{eff}$ (µm) using Eqn. 5 | 33.33 | 25.14 | 18.56 | 15.42 | 14.0 | 21.29±7.98 |
| $\sigma_{ext}$ (km$^{-1}$) from COBALD | 0.013 | 0.018 | 0.017 | 0.018 | 0.026 | 0.018±0.005 |
| *$\sigma_{ext}$ (km$^{-1}$) using Eqn. 6* | *0.014* | *-* | *0.029* | *0.032* | *0.035* | *0.028±0.009* |
| $\beta_{p532}$ (x 10$^{-4}$ km$^{-1}$sr$^{-1}$) | 5.77 | 8.13 | 7.71 | 8.33 | 12.06 | 8.40±2.28 |
| Lidar ratio (sr) | 24.17 | - | 37.44 | 38.04 | 29.05 | 32.18±6.73 |
| IWC (mg m$^{-3}$) using Eqn. 7 | 0.072 | 0.099 | 0.093 | 0.098 | 0.140 | 0.100±0.025 |
| IWC (mg m$^{-3}$) using Eqn. 8 | 0.130 | 0.137 | 0.096 | 0.087 | 0.114 | 0.112±0.021 |
| *IWC (mg m$^{-3}$) from the size distribution* | *0.070* | *-* | *0.157* | *0.195* | *0.285* | *0.176±0.089* |

**3.4 Investigation of the mechanisms involved in the formation of the tropopause cirrus**
**3.4.1 Back-trajectories and their intersection with deep convective clouds observed from Himawari-8**
Overshooting convection can directly inject ice crystals into the lowermost stratosphere region (Corti et al., 2008;
Khaykin et al., 2009; Dessler et al., 2016; Smith et al., 2017; Lee et al., 2019; Khaykin et al., 2022). Under the
subsaturated conditions of the stratosphere, these ice crystals subsequently undergo sedimentation and sublimation
leading to a hydration patch (Jensen et al., 2020; Lee et al., 2019). This hydration patch could get advected and
further lead to secondary ice formation upon cooling near the CPT region. Extreme convective clouds (cloud top
heights exceeding 17 km) can also induce cooling near the tropopause (Kim et al., 2018) which eventually could
cause dehydration through ice formation. In addition to this, rapid tropopause cooling can also be caused by the
convectively induced gravity waves (Kim and Alexander, 2015; Kim et al., 2016) followed by supersaturation, cloud



formation and dehydration (Dzambo et al., 2019; Schoeberl et al., 2015, 2016). A recent study (Dzambo et al., 2019)
over Australia has shown that the magnitude of tropopause cooling is greater during the monsoon period than that
during the non-monsoon periods. CL5 could possibly be a result of the direct injection of ice crystals due to
overshooting convection or through cooling caused by one of these processes mentioned above.

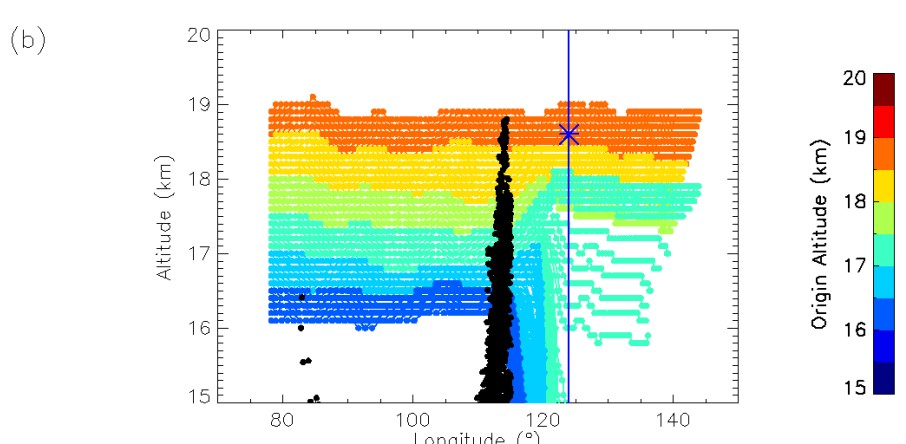

**Figure 5: (a) Five-day back-trajectories initialized on 23 August 2017 at 20:00 UTC from the balloon**
**measurement locations at different altitudes between 16 and 19 km represented by colored dots as a function**
**of latitude and longitude with colour showing the day fraction. The locations of deep convective anvil tops**
**observed after 21 August 2017 from the Himawari-8 brightness temperature images at 10.4 μm wavelength**
**channel that intersected the back-trajectories are shown by black dots. Typhoon *Hato* track is shown by**
**black outlined coloured circles connected by black line with colour showing the day fraction. The night-time**
**CALIPSO orbit track on 20 August 2017 between 17:27 and 17:40 UTC is shown by cyan line with the**
**location of the overshooting cloud top shown by red asterisk. (b) Five day back-trajectories represented as a**
**function of longitude and altitude with colour showing their origin altitude. The locations of the anvil top**
**altitude that intersected with the back-trajectories are represented by black dots. The locations of CALIPSO**
**overpass and the overshooting cloud top altitude are shown by blue vertical line and blue asterisk,**
**respectively.**



1       The influence of deep convection on the CL5 is investigated by using a combination of back-trajectory
analysis and cloud-top brightness temperature observations at 10.4 µm from the Himawari-8 geostationary satellite
data as discussed in Sect. 2.4.1. Figure 5(a) shows five-day back-trajectories initialized from the balloon
measurement sites between 16 and 19 km as a function of latitude, longitude, and time. The locations of the
convective anvil tops that intersected these air parcels are also superimposed along the back-trajectories. Few
isolated localized convective clouds over the Indian East Coast along the back-trajectories can be noticed in Fig.
5(a). While intersecting with the back-trajectories, the anvil top of these convective events does not reach 18 km
altitude. But comparisons of GOES-16 (nearly identical to Himawari-8) and Visible Infrared Imaging Radiometer
Suite (VIIRS) temperature (Khlopenkov et al., 2021) show that pixel resolution has a strong impact on observed
temperatures within convection. Therefore, the highest overshooting tops likely exceeded 18 km height. The
convective clouds near the Indian East Coast were also observed by the India Meteorological Department Doppler
Weather Radars located at Machilipatnam (16.12° N, 81.09° E) after 12:00 UTC on 23 August 2017 with maximum
echo altitude reaching about17 km (Fig. S3). These convective clouds were at their peak altitudes at around 11:40
UTC as seen in the Himawari-8 images, and we do not have radar images prior to 12:00 UTC or co-located
CATS/CALIOP observations to confirm this. However, the intersection of air parcels with these convective clouds
over the Indian East Coast took place nearly 6 hours later at around 16:00 UTC during which their anvil top altitude
reduced. Based on the Himawari-8 images and back-trajectories, the direct injection of ice crystals by these local
convective clouds up to 18 km altitude along the Indian East Coast seems likely. Thus, we cannot fully rule out the
influence of local convection because we notice enhanced WVMR at 70 hPa pressure level on 23 August 2017 at
12:00 UTC over the surrounding regions of Hyderabad and the east Coast (see Fig. S3). This enhanced water vapour
might be associated with these convective clouds as far as the recent studies on the convective hydration are
concerned (Jensen et al., 2020; Ueyama et al., 2018). Enhanced water vapour might have led to the secondary ice
formation after undergoing through cold tropopause conditions while subsequent advection towards Hyderabad by
the TEJ. Ueyama et al. (2018) showed that the deep convective clouds occurring above 380 K potential temperature
level (~17 km) increase in situ formed cirrus clouds by 38 % compared to 21 % by the convective clouds in the
range 375-380 K. This could be the case for CL5.

27       In addition to the hydration, these convective clouds could have also caused cooling near the CPT as
discussed by Kim et al. (2018), which could have facilitated the in situ formation of ice crystals. Temperature
measurements on 23 August 2017 at different times in and around 400 km radius of TIFR-BF suggest that such
cooling has most likely taken place before the arrival of the air parcels as shown in Table 2. Lack of accurate RHi
and ice measurements near the CPT over these convective clouds preclude us from confirming in situ formation.
Thus, the role of local deep convective clouds in the formation of CL5 is not quite clear due to the lack of
observations. High-resolution model simulations of these convective clouds could help in ascertaining their role in
the formation of tropopause cirrus clouds, which is beyond the scope of this study.





**Table 2: Variation of the CPT temperature and altitude as measured by radiosondes and GNSS-RO in and**
**around Hyderabad during 23 and 24 August 2017.**

| Date/Time | 23 Aug 2017, 00 UTC (Radiosonde) | 23 Aug 2017, ~14:00 UTC (GNSS-RO) | 23 Aug 2017, 20:55 UTC (Radiosonde) | 23 Aug 2017, 21:50 UTC (Radiosonde) | 24 Aug 2017, 00 UTC (Radiosonde) |
|---|---|---|---|---|---|
| **Location** | 17.45° N, 78.46° E | 18.45° N, 81.40° E | 17.44° N, 78.30° E | 17.40° N, 77.36° E | 17.45° N, 78.46° E |
| **Displacement from TIFR-BF (km)** | 13.5 | 332 | 38.9 | 135.5 | 13.5 |
| **CPT altitude (km)** | 17.20 | 17.30 | 17.9 | 17.75 | 17.7 |
| **CPT temperature (°C)** | -77.2 | -82.5 | -86.4 | -86.1 | -81.5 |

In Fig. 5(a), we note that the air parcels have already been influenced by deep convective clouds over the South
China Sea before their intersection with local convective clouds over the Indian East Coast. The altitude distribution
of the back-trajectories shown in Fig. 5(b) suggests that the air parcels initialized from the balloon measurement
altitudes between 16 and 18 km have originated from the lower levels (below 10 km) in the region between 115° E
and 125° E on 22 August 2017. This large-scale upward movement of the air parcels indicates a synoptic scale
convective system, such as a tropical cyclone or a typhoon. Himawari-8 brightness temperature image on 22 August
2017 at 19:00 UTC in Fig. 6 does reveal such synoptic scale system named typhoon *Hato,* a category-3 tropical
cyclone in the South China Sea, which severely affected the coastal cities of Macau, Zhuhai, and Hong Kong in
southern China on 23 August 2017 after its landfall (Li et al., 2018). Typhoon *Hato* intensified to a category-3
typhoon just before its landfall (Pun et al., 2019) and it was one of the strongest typhoons in the last several decades,
which caused widespread flooding in the Pearl River Delta region in southern China, Macau being the hardest hit
city (ESCAP/WMO Typhoon Committee, 2017). The anvil cloud top altitude derived from the Himawari-8 cloud-
top temperature images along the back-trajectories are found above 18.5 km (Fig. 5b) with the highest cloud top
altitude of about 18.8 km observed on 22 August 2017 at 09:10 UTC over 18.85° N, 114.48° E. It appears that
typhoon *Hato* had most likely injected ice crystals in the lower stratosphere as the CPT altitude observed from the
nearest radiosonde temperature profile over Haikou station (20.03° N, 110.35° E) at 12:00 UTC was about 17.4 km
(Fig. S4). Moreover, the tropopause temperature and tropopause altitude might be substantially reduced by the
overshooting convection as pointed out by Sun et al. (2021).



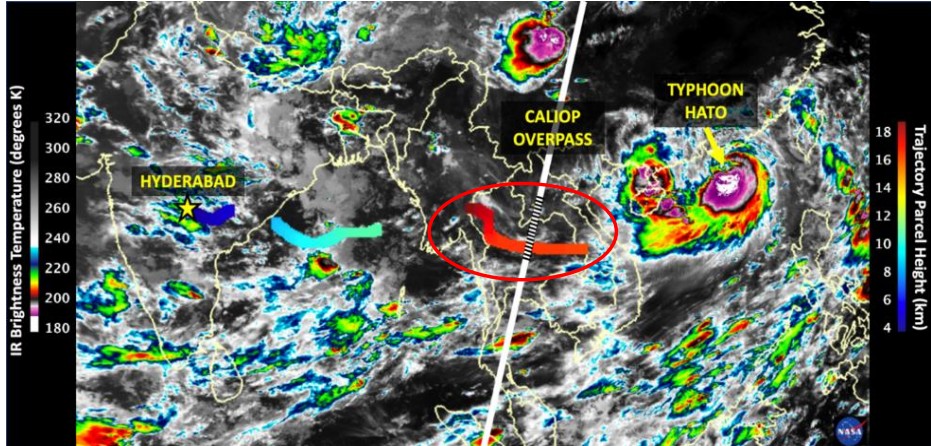

**Figure 6: Image of brightness temperature (in Kelvin) at 10.4 μm infra-red channel of Himawari-8 satellite**
**valid for 22 August 2017 at 19:00 UTC over East Asia. Typhoon *Hato* is associated with very low brightness**
**temperature. The altitudes (in km) of back-trajectory air-masses initialized from the balloon measurement**
**site (Hyderabad, India; denoted as yellow star) are superimposed on the image and are represented by thick**
**coloured curves. The air-mass back-trajectories initialized between 16 and 18 km altitude above Hyderabad**
**at the time of the balloon flight on 23 August 2017 are encircled red. The thick white line shows the night-**
**time CALIPSO overpass from 19:06 to 19:20 UTC on 22 August 2017 with black stripes showing the region**
**where tropopause cirrus was sampled.**
Ice crystals injected by typhoon *Hato* in the subsaturated lower stratosphere might have caused a hydration
patch after undergoing through sedimentation and sublimation while being advected westward by the prevailing
easterly winds. The ERA5 WVMR at 70 hPa pressure level on 22 August 2017 at 09:00 UTC indeed reveals such
hydration patch (see Fig. S4). We did not have collocated CATS/CALIOP observations at that time to verify this
overshooting event. However, there was a nighttime CALIPSO overpass on 20 August 2017 through the developing
typhoon between 17:28 UTC and 17:40 UTC (a few hours before the passage of the air parcels) that observed cloud
top altitude of about 18.6 km (see Fig. 5a, 5b and Fig. S5). Daytime and co-located night-time profiles of the
WVMR from the MLS instrument confirm the presence of such hydration over typhoon *Hato* with peak WVMR
reaching ~6.5 ppmv at 82.5 hPa pressure level well above the CPT observed from the nearest GNSS-RO
temperature profile at 22:44 UTC as shown in Fig. 7. This WVMR value could be even higher after *Hato* intensified
to a category-3 typhoon because intensification is followed by frequent long-lasting intense convective
bursts(Horinouchi et al., 2020) that increase the possibility of overshooting and direct hydration in the lower
stratosphere (Romps and Kuang, 2009; Jiang et al., 2020). Aircraft observations over the Western Pacific region
during the NASA's POSIDON campaign have provided evidence for such direct hydration in the outskirts of
typhoon *Haima* on 15 October 2016 (Jensen et al., 2020). They observed a layer (~1 km thick) of enhanced water
vapour in the lower stratosphere with a peak WVMR of about 7 ppmv at an altitude of ~17.5 km, which could not be
resolved by MLS owing to its coarse vertical resolution (see Fig. 2 of Jensen et al. (2020)).

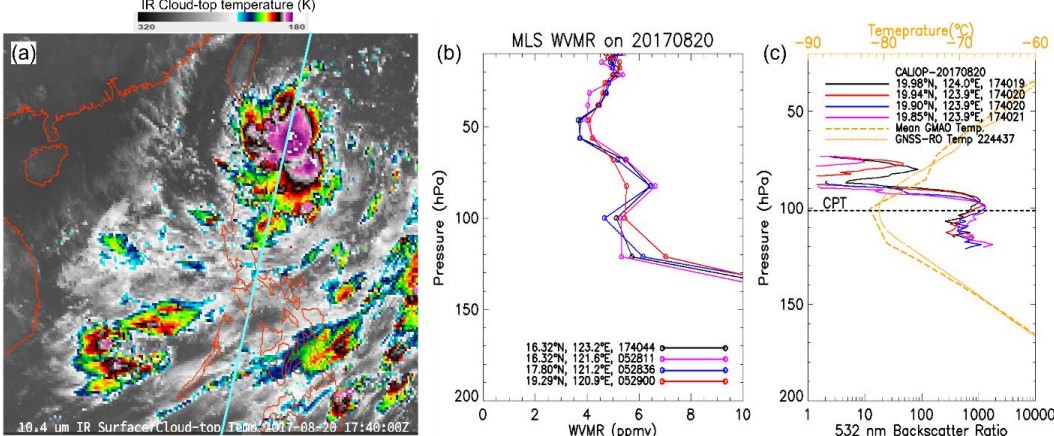

**Figure 7: (a) 10.4 μm infrared cloud-top temperature from Himawari-8 at 17:40 UTC on 20 August 2017 showing the overshoots during the development phase of typhoon *Hato*. The Cyan line represents the CALIPSO orbit track that observed the overshoot on 20 August 2017 between 17:27 and 17:40 UTC. (b) Co-located vertical profiles of water vapour mixing ratio (WVMR) from MLS on 20 August 2017 near the overshoot region observed during daytime and night-time overpasses of Aura satellite. (c) Vertical profiles of 532 nm backscatter ratio obtained from the CALIOP on 20 August 2017 at around 17:40 UTC. Dashed orange lines show the mean temperature profile obtained from GMAO along CALIPSO track and from GNSS-RO at 22:44 UTC. Dashed horizontal black line represents the CPT.**

Jiang et al. (2020) have studied the impact of 30 tropical cyclones (TC, many of them observed during the ASM) on the UTLS water vapour over the tropical northwestern Pacific Ocean using MLS and CloudSat observations during 2012-2016. They found that the lower-stratospheric water vapour over the TC area increased by an average value of 0.75 ppmv compared to the non-TC area, indicating direct hydration. It seems that the hydration caused by typhoon *Hato* has followed the anticyclonic flow and may have influenced the formation of CL5. We investigate this advection with further water vapour observations from MLS and ERA5 derived WVMR in the next sub-section.

**3.4.2 Advection of injected ice and hydration caused by typhoon *Hato***

As discussed in the previous section, the trajectories and the Himawari-8 cloud-top images suggest that the ice crystals detected by our balloon measurements are likely influenced by the convective outflow of typhoon *Hato*, which could possibly be advected to the measurement site by the Tropical Easterly Jet (TEJ or formed in situ from the moist air-mass (from sublimated overshoot) brought by TEJ near the cold tropopause. We discuss these aspects in this section. TEJ is known to play a significant role in the redistribution of upper-tropospheric moisture and the formation of cirrus clouds during the ASM (Das et al., 2011). However, it can be questioned that the ice crystals sampled by our balloon measurements can survive that long distance (about 4000 km) if they were advected from the outflow of a typhoon because larger particles would eventually sediment and sublimate in a subsaturated region. Assuming the ice crystals to be spheroids (aspect ratio=1), the terminal velocity calculated by using the relationship given by Heymsfield and Westbrook (2010) suggests that both 5 and 10 μm particles will fall less than 1 km in two





days. Thus, they are expected to survive within the layer if they were advected from the outflow of typhoon *Hato* by
the TEJ provided they do not grow, or sublimate and the background conditions remain the same. These theoretical
calculations are consistent with the Boulder Counter measurements which show the dominance of ice crystals
smaller than 25 µm in CL5 (Fig. 2c). The presence of ice crystals larger than 10 µm in the middle of the CL5 could
be explained by the constant growth, aggregation, and consequent sedimentation within the layer in the outflow.

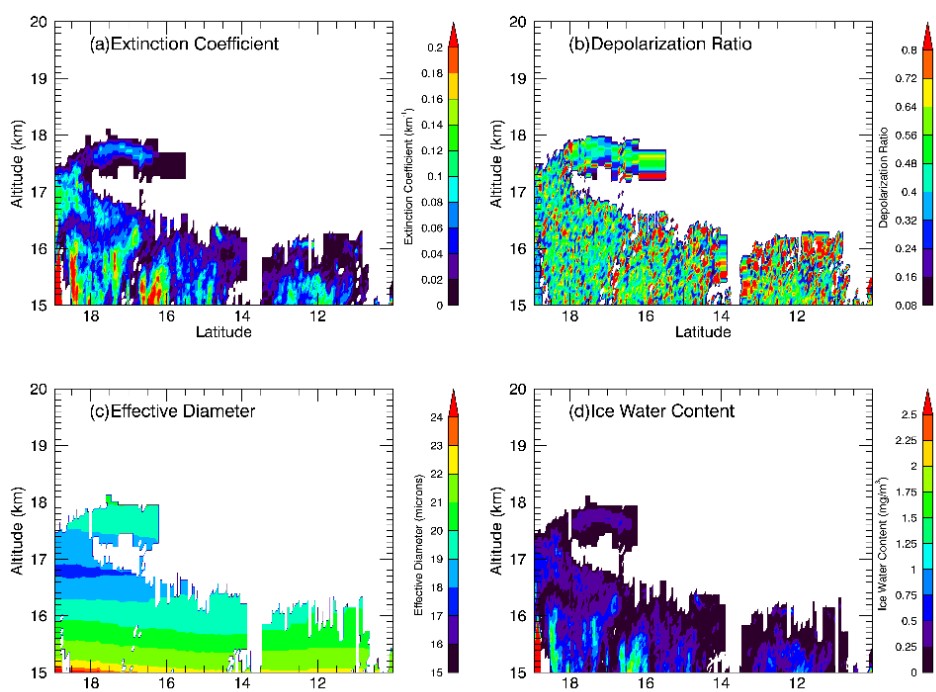

**Figure 8: Optical and microphysical properties of cirrus clouds observed by CALIOP on 22 August 2017 at**
**19:00 UTC over Southeast Asia represented by black stripes in Fig. 6.**

11       A thin cirrus cloud layer can be seen near the tropopause between 17 and 18 km over Laos and Thailand in

the night-time observations from CALIOP on 22 August 2017 at around 19:00 UTC along the air parcels coming
from the typhoon with properties like CL5 (Fig. 8a-d). This cirrus cloud layer extended horizontally to more than
200 km along the CALIPSO orbit track in the same latitude band (17° N - 18° N) where the balloon measurements
were taken. The values of depolarization ratio between 0.3 and 0.6 and colour ratio close to 1 (Fig. S6) in this layer
confirm the presence of non-spherical ice crystals. The CALIOP IWC to extinction coefficient ratio provides an
estimated effective diameter ranging between 18 and 20 µm, which is consistent with our balloon measurements
(20.22 ± 4.54 µm). The peak extinction coefficient value lies between 0.08 and 0.1 km$^{-1}$ and is in the altitude range
of 17.6-17.9 km, consistent with the extinction coefficient maxima (0.06 km$^{-1}$ at 17.9 km from COBALD) for CL5.
The IWC estimated by CALIOP for this layer using an empirical parametrization developed from aircraft data
(Heymsfield et al., 2014) ranges between 0 and 0.5 mg m$^{-3}$. These similarities in optical and microphysical properties



suggest that the two layers detected by balloon and CALIOP could possibly be formed by the same mechanism
under similar background conditions. However, the lifetime of ice crystals in cirrus clouds depend on several local
background factors such as temperature, relative humidity with respect to ice, horizontal wind and updraft speed,
and the type and quantity of ice-nuclei. These factors may vary rapidly especially in the TTL under the influence of
deep convective clouds, TEJ, gravity waves, and relatively high aerosol concentration during the ASM, which could
influence the ice microphysics. This means that the ice crystals formed at one location through one mechanism may
sublimate and/or crystalize at other locations while moist air is being transported to another location depending upon
the local background conditions which force its distribution between ice and gas phases. It is therefore important to
keep a track record of these parameters along the back-trajectories. We discuss the role of background conditions
such as temperature, water vapour and vertical wind speed on the in situ formation of tropopause cirrus clouds along
the back-trajectories in the next sub-section.

12          As mentioned earlier, the convectively injected ice by typhoon *Hato* in the subsaturated lower stratosphere
may undergo sedimentation and sublimation leading to a hydration patch in the lower stratosphere (Lee et al., 2019).
To investigate this, we derive WVMR from ERA5 relative humidity and temperature hourly data at 70 hPa over
typhoon *Hato* on 22 August 2017 at 09:00 UTC where the peak overshoot was observed from the combined analysis
of Himawari-8 and the back-trajectories (Fig. 9a). We see an enhancement in the WVMR located between 15°-20°
N and 107°-114° E. As we move forward in time, this enhanced moist plume is seen to be advected towards the
west. However, a weaker enhancement in WVMR can also be seen over the Indian East Coast surrounding
Hyderabad, probably due to afternoon deep convective clouds, which likely have also assisted the formation of CL5.
The advection of moist plume can also be seen in the averaged WVMR at 70 hPa level for 22 and 23 August 2017
with high WVMR between 15° N and 20° N latitudes as shown in Fig. 9(b).

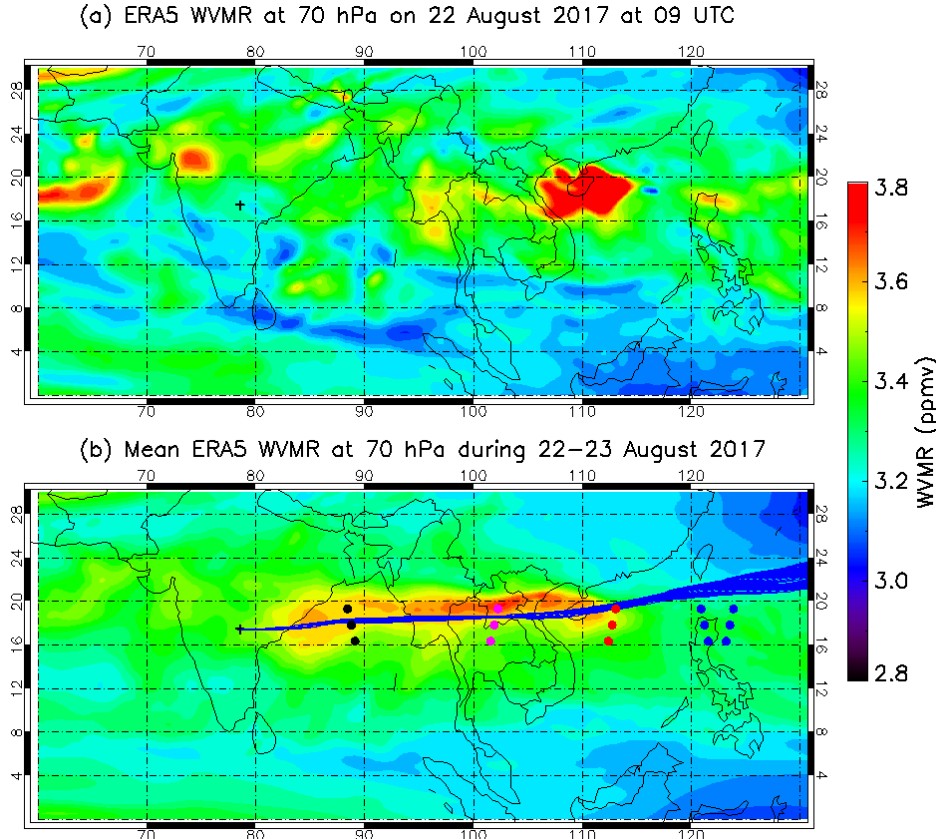

**Figure 9: (a) Spatial map of water vapour mixing ratio (WVMR) derived from ERA5 data at 70 hPa level on 22 August 2017 at 09:00 UTC showing large enhancement over the South China Sea. (b) Average water vapour mixing ratio derived from ERA5 for 22 and 23 August 2017 at 70 hPa with back-trajectories initialized between 18 and 19 km (between 79 and 68 hPa) from the balloon site represented by blue lines superimposed on it. Nearest MLS footprints with respect to the back-trajectories between 16° and 20° N on 20, 21, 22 and 23 August 2017 are represented by blue, red, magenta, and black filled circles, respectively.**

To quantify the transport of this enhanced moisture relative to the monthly mean WVMR at 70 hPa and non-TC area, we estimated WVMR anomaly for the month of August 2017 by subtracting the mean WVMR for each grid over this region (between 15° N and 20° N). The Hovmöller plot in Fig. 10 (a) clearly shows the transport of enhanced moisture (0.6-0.9 ppmv) relative to the monthly mean and reaching Hyderabad on 23-24 August night. We also notice that the magnitude of this enhanced moisture is in good agreement with the results found by Jiang et al. (2020) using MLS observations. We also verified this transport with the MLS WVMR observations during 20 and 23 August 2017 over this region (between 15° N and 20° N) that transacted the back-trajectories nearest in space and time (Fig. 10b). We clearly see enhanced moisture (greater than the monthly mean value) at 82.5 hPa and 68 hPa pressure levels whose magnitude decreases as we move forward in time (from Fig. 10e to 10b). While being advected, this moist plume may get transported downward towards the CPT and may lead to in situ secondary ice



1 formation upon cooling. We present the evidence for such downward transport and cooling near the tropopause

2 along the back-trajectories in section below.

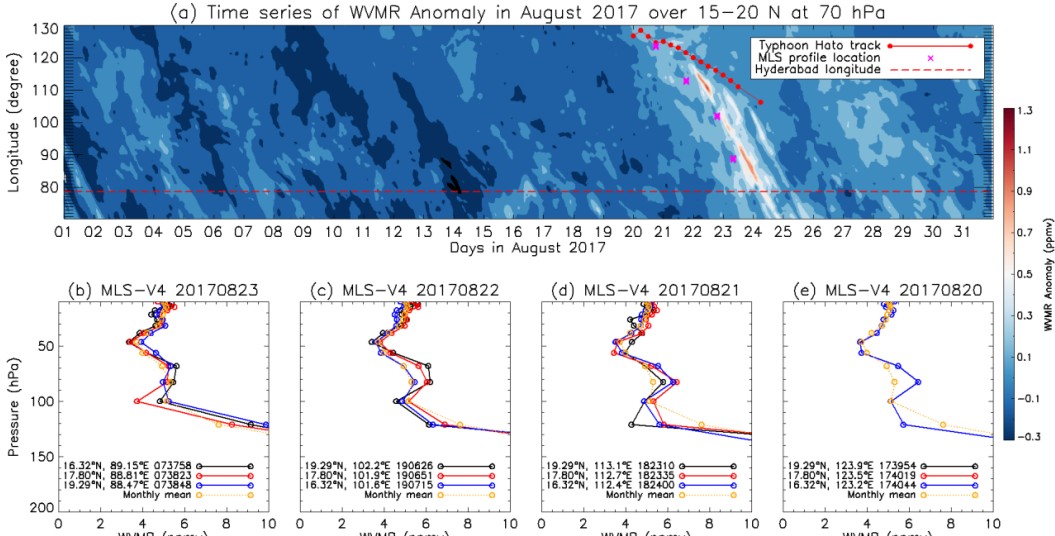

**Figure 10: (a) Hovmöller plot for the anomaly in water vapour mixing ratio (WVMR) derived from ERA5 temperature and relative humidity at 70 hPa pressure level for August 2017. Anomaly is computed by subtracting the monthly mean WVMR for each grid between 15° N and 20° N. Dashed red line marks the longitude of TIFR-BF in Hyderabad. Red dots represent the track of typhoon *Hato* while magenta crosses show the location of nearest MLS profiles on different dates. Vertical profiles of WVMR from MLS between 15° N and 20° N observed on (b) 23 August 2017, (c) 22 August 2017, (d) 21 August 2017, and (e) 20 August 2017 at different times in UTC shown by different colors in the legend. The profiles affected by clouds are filtered out. The orange line with orange circles in each panel represents the mean MLS profile for August 2017 obtained by averaging all profiles located in the region bounded by 15°-20° N and 70°-130° E.**

**3.4.3 In situ ice formation due to cooling induced by gravity waves associated with typhoon *Hato*.**
Figure 11 shows the temperature history of the air parcels along the back-trajectories initialized from the balloon
measurement site between 16 and 19 km altitude every hour. After being influenced by typhoon *Hato,* the air parcels
have experienced several cooling and warming phases before reaching the balloon site. We observe a quasi-periodic
occurrence of cold regions with temperature below -81 °C along the back-trajectories after being influenced by
typhoon. Colder air parcels at these temperatures are susceptible to generate supersaturation that can trigger the
formation of ice crystals through ice nucleation. Near coincident and co-located CATS and CALIOP observations
on 22 and 23 August 2017, which intersected the back-trajectories, are used to investigate the presence of cirrus
clouds in these cold-regions. CATS and CALIOP observations on 23 August 2017 are almost coincident with the air
parcels along the back-trajectories whereas CALIOP overpass on 22 August 2017 was a few hours later to the
passage of the air parcels and thus, there is a difference in time (Fig. S7). It is interesting to see the presence of
cirrus cloud layers with their base altitude greater than 16 km in and around these cold regions along the CALIPSO
and CATS tracks.



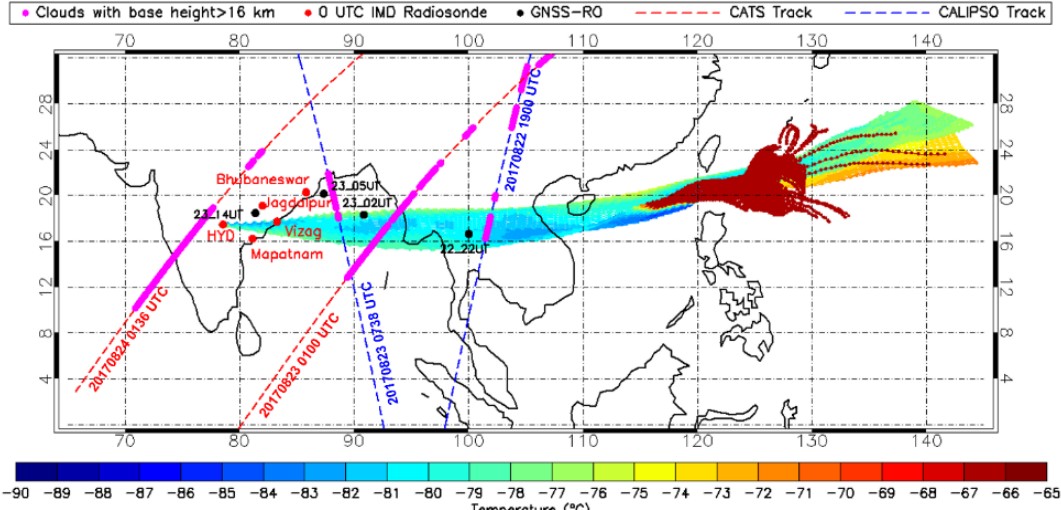

Figure 11: Filled coloured circles on the map show the temperature and location of the air parcel at every hour along the back-trajectories run from the balloon measurement sites between 16 and 19 km on 23 August 2017 at 20:00 UTC. Red and blue dashed lines represent the nearest orbit tracks of CATS (on 24 August 2017 at 01:36 UTC and 23 August 2017 at 01:00 UTC) and CALIPSO (23 August 2017 at 07:38 UTC and 22 August 2017 at 19:00 UTC), respectively, with respect to the air parcels. Filled magenta circles superimposed on the orbit tracks represent the locations of cirrus cloud layer with base altitude greater than 16 km. Red filled circles on the map show the locations of the IMD stations from where daily radiosondes are launched at 00 UTC. Black filled circles show the locations of available GNSS-RO temperature profiles closest in time and space with CATS and CALIPSO overpasses.

The periodicity and the vertical extent of these cold regions can be clearly seen in Fig. 12 (a) where hourly temperature along the back-trajectories initialized from the balloon site between 16 and 19 km altitude at every 100 m has been shown as a function of altitude (and potential temperature, see Fig. S8). The periodic cold regions are confined in the altitude range between 16.5 and 18 km (potential temperature between 370 K and 400 K, see Fig. S8) containing the CPT. The anvil top height derived from Himawari-8 cloud-top temperature, which intersected the back-trajectories initialized between 16 and 19 km, is also superimposed (Fig. 12 a). The coldest temperatures are found over the strongest updraft regions of typhoon *Hato* with highest cloud tops reaching ~18.5 km. The occurrence of cirrus clouds between 16 and 18 km altitude in these cold regions is also clear from the near co-located and coincident measurements from CATS and CALIOP (Fig. 12 a). We observe westward propagating wave-like patterns in the temperature of the air parcels after they are influenced by typhoon *Hato*. There is also an upward movement of these wave-patterns, which are more prominent near 18 km and altitudes above originating right from the region of typhoon *Hato*. Such periodic temperature fluctuations near the tropopause have also been observed along the diabatic back-trajectories (derived from ERA5 data) after being influenced by a typhoon observed during the ASM in a recent study (Li et al., 2020). Li et al. (2020) investigated the influence of two typhoons on the dehydration and transport of low-ozone air masses near the tropopause during the ASM over Kunming, China using balloon measurements of temperature, ozone, and water vapour. However, such wave-like patterns in temperature and their role in cirrus cloud formation along the back-trajectories have not been discussed in



their study. These wave-like temperature fluctuations or temperature anomalies near the tropopause could be due to
the gravity waves generated by the deep convection associated with typhoon *Hato*.

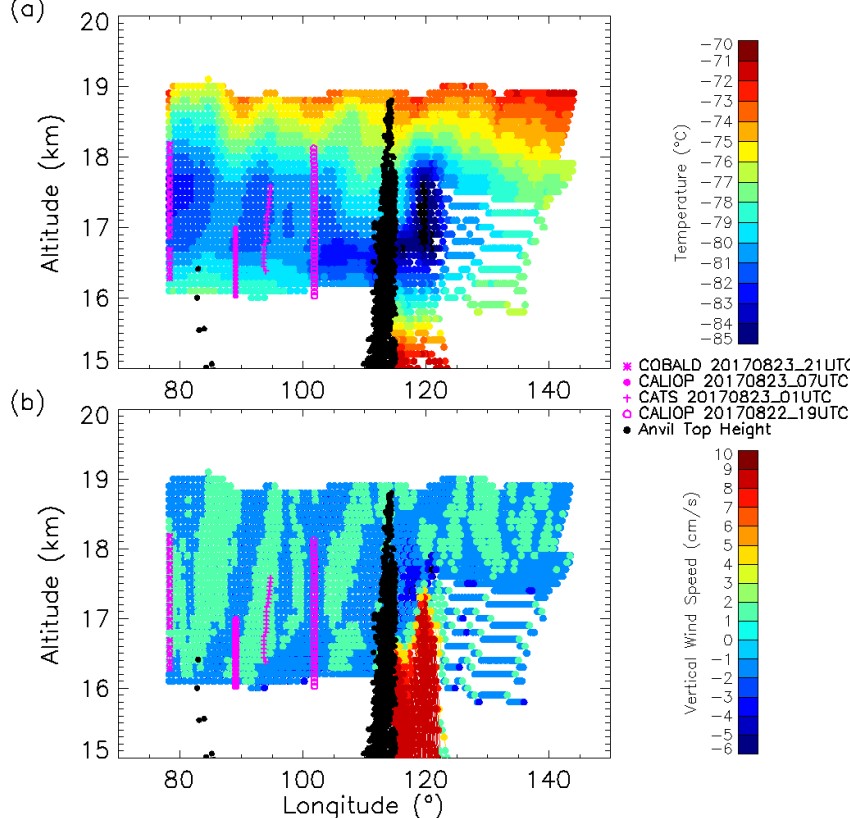

**Figure 12: (a) Temperature history of the air parcels along the back-trajectories run from the balloon**
**measurement sites between 16 and 19 km on 23 August 2017. Each filled coloured circle represents**
**temperature value at every 100 m vertical resolution at each hour. (b) Same as (top) but for the vertical wind**
**derived from the location and time of the back-trajectories. Different magenta symbols represent the location**
**of the layered cirrus clouds observed near the tropopause from CALIOP, CATS and balloon measurements**
**near co-located in space and time with respect to the back-trajectories. The locations of anvil cloud top**
**altitude are represented by black dots.**
Using the radiosonde observations over the tropical Western Pacific region, Kim and Alexander (2015)
have shown that the CPT temperature is directly modulated by the vertically propagating waves irrespective of any
change in the mean upwelling. Later, Kim et al. (2016) showed ubiquitous influence of waves on the TTL cirrus
clouds using airborne observations of temperature and cirrus clouds obtained over the Western Pacific during the
ATTREX campaign. Several studies (Wu et al., 2015; Nolan and Zhang, 2017; Kim et al., 2009) have reported
anomalies in temperature and vertical wind speed associated with semi-circular gravity waves generated from the
typhoon centre propagating horizontally and vertically through the troposphere, stratosphere, and mesosphere in
expanding spirals with horizontal wavelengths typically in the range of 50-500 km and periods from 1 hour to 1.6
days. We derived vertical wind speed along the back-trajectories (Fig. 12b) which shows correspondence between



the periodic updrafts and cold anomalies. This suggests that the areas of cold anomalies are most likely formed due
to the cooling induced by the updraft. It has been found that the updraft motion slows down the process of
sedimentation, leading to longer lifetime of ice crystals in cirrus clouds (Podglajen et al., 2018). The downdrafts
along the back-trajectories might have brought the moisture from the lower stratospheric hydration patch towards
the CPT, resulting in secondary ice formation. This is evident from Fig. 5(b) where air parcels that originated
between 17 and 17.5 km can be seen descending from higher altitudes at around 120° N. One more interesting point
to note is that the ascent speed of the balloon has increased from ~6 m s$^{-1}$ to ~8 m s$^{-1}$ near the tropopause region
(17.5-18.5 km) with its peak value at the top of CL5, above which it decreases sharply (see Fig. S9) and beyond that
it exhibits oscillatory behaviour with increasing amplitude. It is interesting to note the presence of downdrafts (Fig.
12b) and sudden drop in balloon ascent speed above the CL5 top. Such vertical wind shear near the CPT could
influence the ice nucleation with sudden rising motion triggering it while sinking motion inhibits it. The association
of cold anomalies with updrafts indicate towards the possible role of gravity wave influence, which we investigate
below.
The temperature along the back-trajectories in our analysis is obtained from the MERRA-2 reanalysis data,
which may not be accurate enough to resolve these temperature fluctuations near the CPT when compared with the
observations (Tegtmeier et al., 2020). To confirm the robustness of these cold anomalies, we use high resolution and
relatively more accurate temperature observations from GNSS-RO near these colder regions. We found temperature
profiles within 6 hours interval and within 400 km radius from the location of mean scattering ratio profiles of
CATS and CALIOP intersecting the back-trajectories between 16° N and 18° N latitude band as shown in Fig. 13
(b), (c) and (d) arranged according to their observation time (from latest to farthest). The temperature near the
tropopause is better resolved in the GNSS-RO temperature profiles as compared to the GMAO temperature profile
averaged between 16° N and 18° N latitude band along the CATS and CALIOP orbit track. It is clear from the
scattering ratio profiles obtained from the COBALD (at 940 nm, Fig. 13 a), CALIOP (at 532 nm, Fig. 13 b and d)
and CATS (at 1064 nm, Fig. 13 c) that layered cirrus clouds or laminar cirrus are observed near the CPT in these
cold anomalies. Profiles of RHi observed from the Aura-MLS on 22 and 23 August 2017 co-located with CALIOP
show RHi values greater than 100 % at 100 hPa (~16.8 km) level, indicating supersaturation that resulted in ice
formation (see Fig. S10).

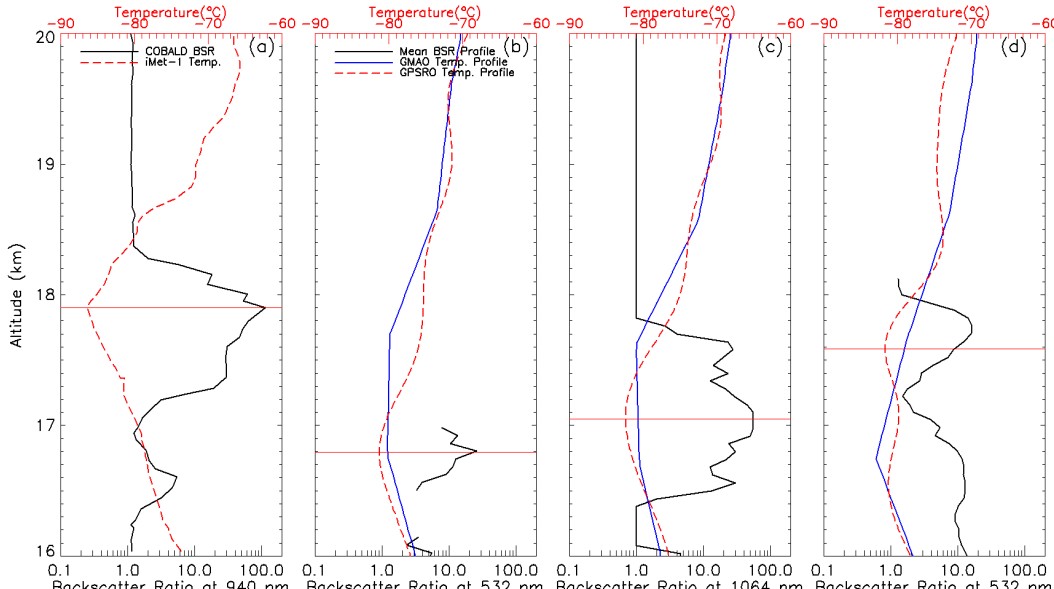

**Figure 13: Vertical profile of scattering ratio (shown by black line) obtained from (a) COBALD data on 23**
**August 2017 at around 20:40 UTC over 17.45° N, 78.31° E, (b) CALIOP on 23 August 2017 at around 07:38**
**UTC over 16.99° N, 88.88° E, (c) CATS on 23 August 2017 at around 00:59 UTC over 17.01° N, 92.68° E and**
**(d) CALIOP on 22 August 2017 at around 19:07 UTC over 16.99° N, 101.68° E. CALIOP and CATS profiles**
**intersecting with the back-trajectories are averaged between 16° and 18° N. Blue line in (b), (c) and (d)**
**represents the mean temperature profile obtained by averaging the corresponding MERRA-2 temperature**
**profiles provided with CALIOP and CATS data. Dashed red line shows the co-incident and co-located**
**temperature profile obtained from (a) radiosonde measurement, (b) COSMIC GPS RO on 23 August 2017 at**
**05:07 UTC over 20.13° N, 87.37° E, (c) COSMIC GPS RO on 23 August 2017 at 02:33 UTC over 18.29° N,**
**90.87° E, (d) COSMIC GPS RO on 22 August 2017 at 23:42 UTC over 16.63° N, 100.04° E. Horizontal red**
**line in each panel marks the CPT.**
The CPT temperature and CPT altitude over Hyderabad (within 400 km radius around TIFR-BF) on 23
August 2017 exhibited a drastic change in the last 24 hours as observed from balloon and GNSS-RO observations as
discussed earlier (in Table 2). Deep convective clouds that occurred over the land along the Indian East Coast in the
late evening hours (as mentioned in Sect. 3.4.1) might also have helped in strengthening this tropopause cooling.
CPT temperature was minimum and CPT altitude was maximum for this BATAL flight. All the IMD stations near



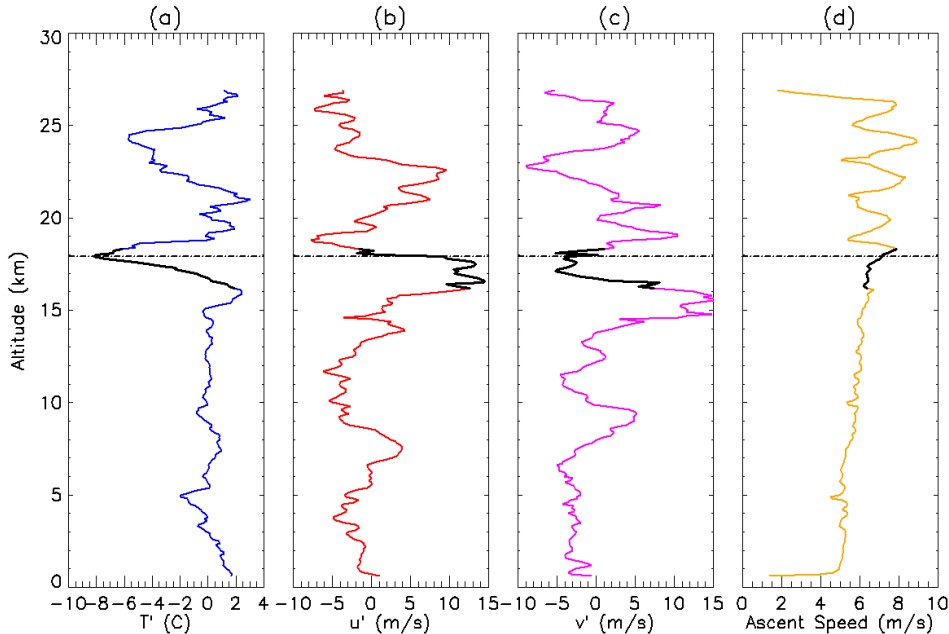

**Figure 14: Vertical profile of (a) temperature anomaly, (b) zonal wind speed anomaly, (c) meridional wind**
**speed anomaly and (d) ascent speed of the balloon observed on 23 August 2017. Anomalies are estimated by**
**subtracting the monthly mean profile from the profile obtained on 23 August 2017. Mean profiles are**
**obtained from all the radiosonde profiles at 00 UTC and BATAL flights over Hyderabad for August 2017.**
**The black dashed horizontal line in each panel denotes the location of the CPT (at 17.9 km) observed on 23**
**August 2017 at around 21:00 UTC from the balloon. Black shading in each profile shows the location of the**
**tropopause cirrus cloud.**
Hyderabad (Machilipatnam, Vizag, and Jagdalpur) recorded lower tropopause temperature on 24 August 2017 at 00
UTC compared to that observed on 23 August 2017 at 00 UTC. The magnitude of tropopause cooling can be
expressed in terms of anomaly in the observed temperature with respect to the mean observed temperature over
Hyderabad. We construct a mean temperature profile over Hyderabad region during August 2017 using the
temperature profiles obtained from all (nine) the flights of BATAL-2017 campaign over TIFR-BF and daily 00 UTC
radiosonde data from Hyderabad airport. The vertical profile of the observed temperature anomaly for our balloon
flight on 23 August 2017 estimated after subtracting the mean temperature profile from it clearly shows a cold
anomaly of ~ -8 °C) near the CPT region (Fig. 14 a) where CL5 was found. We also notice a wave-like pattern in the
temperature anomaly profile above 15 km altitude. Using the airborne data from the ATTREX campaign, Kim et al.
(2016) showed that about 86% of the cirrus clouds between 16.5 and 18 km over the Pacific are formed in the cold
anomalies induced by gravity waves. They also found that these clouds more often form in the negative slope of the
temperature anomaly, which is also true for CL5 (see the cloud shading in Fig. 14) but it is also present in the
positive anomaly above the tropopause as well up to the CL5 top altitude where ascent speed peaks and drops (see
Fig. 14d). Similar wave like pattern is also noticed in the anomaly profiles of zonal and meridional wind speed (Fig.
14b and 14c), estimated by subtracting the mean zonal and meridional speeds for August 2017 obtained from



BATAL flights and Hyderabad airport radiosonde data for August 2017. Such wave pattern is also reflected in the
spatial movement of the balloon as shown in Fig. S1. In addition to this, the ascent speed of the balloon also shows
oscillation above the CPT with its amplitude increasing with altitude (Fig. 14d). All these observations indicate
towards the possible influence of gravity waves in the tropopause cooling, which possibly led to ice formation after
reaching supersaturation with respect to ice. Recently, Martínez et al. (2021) have reported a case of extremely thin
cirrus cloud layer formed near the CPT over the Southwestern Indian Ocean using COBALD and CFH observations.
Using the spectral analysis (S-transform, Stockwell et al. (1996)) of temperature anomaly and ascent speed, they
suggested the role of homogeneous freezing under the influence of a high frequency gravity wave with a vertical
wavelength of 1.5 km. Using Hodograph analysis of radiosonde data following Leena et al. (2012), we estimated the
gravity wave characteristics. We found that the gravity wave was propagating from south-east direction to north-
west direction in the stratosphere (18-26 km) with a horizontal phase speed of about 18.8 m s$^{-1}$ and a horizontal
wavelength of about 1770 km, confirming that it was generated from typhoon *Hato*.
**4 Summary**
In situ measurements in cirrus clouds with ice crystals smaller than 100 μm near the tropopause over the ASM
region are sparse, and the available observations mainly come from probes on aircraft that are traveling at rapid
speeds through the clouds, leading to shattering of ice crystals and failure to resolve small ice crystal habits. In this
paper, we have presented balloon-borne measurements of the optical and microphysical properties of a tropical
tropopause cirrus cloud layer obtained using a backscatter sonde and an optical particle counter on 23 August 2017
during the BATAL campaign over Hyderabad, India. This layer was a subvisible cirrus cloud located at an
extremely cold tropopause temperature of -86.4 ℃. The top of this layer was found in the lower stratosphere at
about 18.3 km. Ice crystals in this cloud layer were smaller than 50 μm in diameter. Nearby lidar backscatter
measurements from the CATS onboard ISS confirmed the presence of this tropopause cirrus that extended more
than 500 km along the ISS orbit track. IWC for this layer was estimated independently from the backscatter
measurements and optical particle counter and found to be less than 0.2 mg m$^{-3}$. Simultaneous measurements of
backscatter coefficient from COBALD and extinction coefficient derived from Boulder Counter measurements
allowed us to derive a range-resolved lidar ratio for this tropopause cirrus cloud layer, whose average value is
32.18±6.73 sr .We demonstrated that the combination of COBALD and Boulder Counter can be used to estimate
range-resolved lidar ratios for tropopause cirrus cloud layers, which may prove useful in validating lidar retrievals of
extinction coefficient. This estimate can be improved with higher ice particle sampling rate at more size channels
especially at smaller radii. We also investigated the formation mechanism of this layer using back-trajectories,
satellite, and ERA5 reanalysis data. Back-trajectories from the balloon measurement site and their intersection with
convective clouds observed in Himawai-8 brightness temperature images suggest that the layer was influenced by a
category-3 typhoon named *Hato*. Both Himawari-8 and CALIOP observations showed that *Hato* injected ice crystals
into the lower stratosphere causing a hydration patch as revealed by ERA5 WVMR and confirmed by the MLS
WVMR observations. These moist plumes were seen to be advected by the ASM anticyclonic flow towards



Hyderabad. Moreover, along the back-trajectories there were quasi-periodic cold and warm anomalies near the CPT
associated with updrafts and downdrafts, respectively. CATS and CALIOP observations showed the presence of
tropopause cirrus clouds in these cold anomalies. These perturbations are most likely caused by the gravity waves
induced by typhoon *Hato*. Signatures of such perturbations were also noticed in the temperature and wind profiles
obtained from our balloon observations. Through this case study, we conclude that the overshooting clouds in
typhoons can cause direct stratospheric hydration during the ASM in addition to the usual overshooting convective
systems. Following the ASM anticyclonic flow, this stratospheric hydration can get advected several thousands of
kilometres and subsequently may lead to tropopause cirrus clouds upon cooling induced by typhoon induced gravity
waves.

10       The occurrence frequency of tropopause-penetrating deep convective clouds (Aumann et al., 2018) and the

intensity of tropical cyclones are expected to increase in a warmer climate (Stocker et al., 2013; Emanuel, 2005),
which in turn are likely to increase the occurrence of ice-injections with consequences for the stratospheric
composition, thin tropopause cirrus clouds and further feedbacks on the global climate (Dessler et al., 2016;
Solomon et al., 2010). A recent modelling study (Smith et al., 2022) has found that the impact of convective
hydration in response to increased $CO_2$ depends not only on the frequency and penetration altitude of convective
overshooting into the stratosphere but also on large-scale temperatures in the TTL. In this context, the occurrence of
frequent deep convection and presence of cold temperatures in the TTL over the ASM anticyclone region are
important for convective hydration and tropopause cirrus cloud formation. In future, simultaneous measurements of
temperature, water vapour, and the microphysical properties of cirrus clouds using quasi-isentropic balloon flights
within the coldest regions of the ASM anticyclone will be planned to obtain a detailed understanding of the impacts
of overshooting convection on the formation of tropical tropopause cirrus clouds.
**Data availability**
CALIPSO data were obtained from NASA Langley Research Centre Atmospheric Science Data Centre
(https://asdc.larc.nasa.gov/project/CALIPSO), GEOS-5 FP wind data from NASA GMAO and Himawari-8 data
from the University of Wisconsin - Madison Space Science and Engineering Centre (SSEC). Doppler weather radar
data were obtained from India Meteorological Department (IMD). Daily 00 UTC radiosonde data were obtained
from University of Wyoming (https://weather.uwyo.edu/upperair/sounding.html), GNSS-RO temperature profile
data from UCAR-COSMIC (https://cdaac-www.cosmic.ucar.edu/), ERA5 reanalysis data from Copernicus climate
data store (https://cds.climate.copernicus.eu/cdsapp#!/dataset/reanalysis-era5-single-levels?tab=overview), Aura-
MLS data from NASA-JPL (https://disc.gsfc.nasa.gov/datasets/ML2RHI_004/summary) and CATS data from
NASA-GSFC (https://cats.gsfc.nasa.gov/). BATAL data will be made available at https://science-
data.larc.nasa.gov/BATAL/data.html and can also be obtained through request.
**Author contribution**
AKP wrote the manuscript draft. JPV, TDF, BSK, HG, MVR, AKP and AJ organized the field campaign. AKP and
JPV collected in situ data. TDF and HG provided back trajectories. AKP and JPV analysed in situ and back-



trajectory data. KMB provided cloud top heights from Himawari-8 along the back-trajectories. MAA contributed to CALIPSO data analysis. AKP analysed CATS, CALIPSO, MLS, GNSS-RO and ERA5 data. FGW provided calibrated COBALD data. SD contributed to ERA5 data analysis. KAJ provided doppler weather radar analysis. MVR provided gravity wave characteristics. All the authors reviewed and edited the manuscript.

**Competing interests**

The authors declare that they have no conflict of interest.

**Acknowledgements**

This work was carried out under ISRO-NASA joint project called BATAL. The 2017 balloon deployment and the analysis of the data were supported by the NASA ROSES Upper Atmospheric Research and the Upper Atmospheric Composition Observations programs through an IDIQ task at NASA Langley Research Centre. AKP is thankful to the Physical Research Laboratory in Ahmedabad, India for supporting his participation in the BATAL-2017 field campaign. He is also thankful for the support provided by the NASA Postdoctoral Program (NPP) fellowship administered by Universities Space Research Association, NASA Langley Research Centre, and National Institute of Aerospace in conducting this research. We thank the engineers, technicians, and staffs at TIFR Balloon Facility in Hyderabad and National Atmospheric Research Laboratory at Gadanki, India for their valuable contributions towards the scientific, technical, and logistical support provided during the BATAL campaign. We also thank Mijeong Park from NCAR for helpful discussion.





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
