# Peer review of "Investigating the role of typhoon-induced gravity waves and"

_EGUsphere, 2023_

## Referee Comment (RC3)

**Review of**
**"Investigating the role of typhoon-induced gravity waves and stratospheric hydration in the formation of tropopause cirrus clouds observed during the 2017 Asian monsoon"**
**by Amit Kumar Pandit et al.**

**1 General comments**

This study reports on a subvisible cirrus cloud observed at the tropical tropopause in one of the BATAL soundings. The instrumentation carried by the balloon on this flight includes a backscatter sonde and a particle counter, besides a standard radiosonde. The study first presents how these instruments are used to precisely characterize the cirrus microphysical properties. In the following part, the authors try to identify the mechanisms that could have led to the formation of this cirrus. Combining backtrajectories with satellite brightness temperature maps and lidar soundings, they suggest that ovsershooting convection associated with typhoon Hato over the South China sea a few days before the cirrus observation may have injected water vapor in the lower stratosphere. The formation of the cirrus was then cause by gravity-wave induced cooling while the water-vapor enriched air parcels were advected toward India by the monsoon anticyclone.

The article is to my opinion a very nice observational study, which provides sound evidence for the formation mechanism that is advocated. The paper is well-written and the argumentation is easy to follow. I would thus recommend its publication with only minor revisions, which are described below.

**2 Minor issues**

- The presentation of Section 2.2.2 (Solair Boulder Counter) may be improved. From line 9 on page 8 to the end of the section, the text does not provide details specifically on the instrument, but rather describes how derived quantities (effective diameter or Ice Water Content) can be inferred from the raw counter or (more confusingly) from the backscatter sonde observations. It may be easier for the reader if an own separate subsection were devoted for this derived quantities.

- Figure 2: It will help the reader to have an additional vertical pressure scale in this figure. Since pressure is very likely measured by the radiosonde, this should not be an issue to add this scale, and it would greatly ease the comparisons with figures 7b, 9 and 10b, which display water vapor on pressure levels.

- Figure 3: I am uncertain about the relevance of the ERA5 cloud cover fraction on the top panel. Since the previous figure showed multiple cloud layers, it is probably quite speculative to make a link between the ERA cloud cover fraction and the CL5 cirrus cloud studied in this paper. The fraction numbers are themselves furthermore very low... On the lower panel of this figure, or on the previous figure, an ERA5 vertical profile of relative humidity over ice might on the other hand provide some additional information.

- p17, l1-2: I am a bit skeptical about the quite optimistic statement that the effective diameter obtained with Eq. (5) is in good agreement with the observations. Indeed, in Table 3, one observes that the effective diameter is monotonically increasing as temperature decreases for the observations (Eq. 3), whereas it is continuously decreasing when estimated according Eq. 5. I have therefore the impression that the claimed agreement is somehow fortuitous here.

- p28, l18-19: actually I do not see the quasi-periodic feature in temperature in Figure 11, but rather in Figure 12.

- p33, l17-18: Be careful though that the 8-10K decrease emphasized in Figure 14 is a Eulerian perturbation. In other words, it is different from the cooling that may have undergone air parcels coming above Hyderabad on that day. The temperature fluctuations felt by air parcels are those shown in Figure 12.

**3   Additional corrections**

- p9, l15: the sentence that starts here should be rephrased.

- Table 1: please use $D_e$ instead of $D_{eff}$, as in the text.

- Table 2: Distance rather than displacement?

- p24, l22: a right ) is lacking after TEJ.

- p27, l5: a space is lacking before 19 km.

- p29, l22: propagation rather than movement.

---

## Author Comment (AC1)

**Response to the review (by Referee #1) of**
**"Investigating the role of typhoon-induced gravity waves and**
**stratospheric hydration in the formation of tropopause cirrus clouds**
**observed during the 2017 Asian monsoon"**
**by Amit Kumar Pandit et al.**

**1 General comments**

This study reports on a subvisible cirrus cloud observed at the tropical tropopause in one of the BATAL soundings. The instrumentation carried by the balloon on this flight includes a backscatter sonde and a particle counter, besides a standard radiosonde. The study first presents how these instruments are used to precisely characterize the cirrus microphysical properties. In the following part, the authors try to identify the mechanisms that could have led to the formation of this cirrus. Combining backtrajectories with satellite brightness temperature maps and lidar soundings, they suggest that ovsershooting convection associated with typhoon Hato over the South China sea a few days before the cirrus observation may have injected water vapor in the lower stratosphere. The formation of the cirrus was then cause by gravity-wave induced cooling while the water-vapor enriched air parcels were advected toward India by the monsoon anticyclone.

The article is to my opinion a very nice observational study, which provides sound evidence for the formation mechanism that is advocated. The paper is well-written and the argumentation is easy to follow. I would thus recommend its publication with only minor revisions, which are described below.

Reply: We thank the referee for going through our manuscript and providing constructive feedback for its further improvement. We are glad to read the referee's views on our manuscript. In the following, our point-by-point response to the referee's comments (in black text) are presented in blue text.

**2 Minor issues**

• The presentation of Section 2.2.2 (Solair Boulder Counter) may be improved. From line 9 on page 8 to the end of the section, the text does not provide details specifically on the instrument, but rather describes how derived quantities (effective diameter or Ice Water Content) can be inferred from the raw counter or (more confusingly) from the backscatter sonde observations. It may be easier for the reader if an own separate subsection were devoted for this derived quantities.

Reply: We are thankful to the referee for pointing this out. As suggested, we have separated Section 2.2.2 into two subsections, one (Section 2.2.2.1) containing the details of Solair Boulder Counter while the other (Section 2.2.2.2) describing the estimation of cloud microphysical properties from COBALD and Boulder Counter data in the revised manuscript.

• Figure 2: It will help the reader to have an additional vertical pressure scale in this figure. Since pressure is very likely measured by the radiosonde, this should not be an issue to add this scale, and it would greatly ease the comparisons with figures 7b, 9 and 10b, which display water vapor on pressure levels.

Reply: We completely agree with the referee and thank him/her for this suggestion. We have added a vertical pressure scale in Fig. 2b, as suggested.

• Figure 3: I am uncertain about the relevance of the ERA5 cloud cover fraction on the top panel. Since the previous figure showed multiple cloud layers, it is probably quite speculative to make a link between the ERA cloud cover fraction and the CL5 cirrus cloud studied in this paper. The fraction numbers are themselves furthermore very low... On the lower panel of this figure, or on the previous figure, an ERA5 vertical profile of relative humidity over ice might on the other hand provide some additional information.

Reply: We agree with the referee and thank him/her for this suggestion. We have replaced ERA5 cloud fraction with ERA5 temperature at 100 hPa as suggested by another referee (Referee#3). We have also added the vertical profiles of relative humidity over ice from ERA5 during the ascent and decent of the balloon in Fig. 3c, as suggested.

• p17, l1-2: I am a bit skeptical about the quite optimistic statement that the effective diameter obtained with Eq. (5) is in good agreement with the observations. Indeed, in Table 3, one observes that the effective diameter is monotonically increasing as temperature decreases for the observations (Eq. 3), whereas it is continuously decreasing when estimated according Eq. 5. I have therefore the impression that the claimed agreement is somehow fortuitous here.

Reply: We agree with referee's argument and therefore, we have deleted that statement in the revised manuscript.

• p28, l18-19: actually I do not see the quasi-periodic feature in temperature in Figure 11, but rather in Figure 12.

Reply: We agree with the referee. The quasi-periodic feature in temperature is not clearly visible in Fig.11 due to the overlap of air parcels originating from different altitudes. In the revised manuscript, we have labelled the quasi-periodic features in Fig.11.

• p33, l17-18: Be careful though that the 8-10K decrease emphasized in Figure 14 is a Eulerian perturbation. In other words, it is different from the cooling that may have undergone air parcels coming above Hyderabad on that day. The temperature fluctuations felt by air parcels are those shown in Figure 12.

Reply: Thanks for this comment. We completely agree with the referee's comment.

**3 Additional corrections**

• p9, l15: the sentence that starts here should be rephrased.

Reply: The sentence is rephrased, as suggested.

• Table 1: please use De instead of Deff , as in the text.

Reply: We have replaced "Deff" with "$D_e$" in Table 1, as suggested.

• Table 2: Distance rather than displacement?

Reply: We have replaced "displacement" with "distance", as suggested.

• p24, l22: a right ) is lacking after TEJ.

Reply: We have added a right ")" after "TEJ".

• p27, l5: a space is lacking before 19 km.

Reply: This has already been corrected.

• p29, l22: propagation rather than movement.

Reply: We have replaced "movement" with "propagation".

---

## Author Comment (AC2)

**Response to Referee's (Referee #2) Comments**

The authors present a study case of the formation and characteristics of a cirrus cloud using the combination of multiple observational tools. For the method, they show how the combination of COBALD and Boulder Counter can be used for the estimation of lidar ratio for optically thin cirrus. Their results highlight the role of gravity waves and crystals injection from the typhoon Hato to explain the formation of the cloud. It describes a new method and an observational study case of the observed impacts of the gravity waves which is significant for the understanding of the interplay between small scale dynamics and microphysics. Finally, it thus also highlights the impacts of typhoons on cirrus clouds.

Although the method is interesting and new to my knowledge, the paper is long and would be improved by a more concise writing to improve the understanding of the methods and key findings of the authors. Therefore I recommend a major revision.

Reply: We thank the referee for going through the manuscript thoroughly and providing constructive suggestions/ideas for its further improvement. Our response to each comment or suggestion (in black text) is shown below in blue text.

Specific comments :

- 1) The paper is quite long and modifications to shorten it a bit might be useful for a better understanding of the study. Here are a few suggestions/ideas to do so :

In the data method part, the authors wrote a short summary of the history/functioning of the instruments COBALD, SOLAIR, CALIOP, CATS, Himawari-8, added to cited literature relative to each one of the instruments (which should be enough). This might be adding details that are not crucial for the understanding of the author's method.

Reply: We appreciate the ideas/suggestions provided by the referee to shorten the data and method section. As suggested, we have tried to shorten the description of data and method section by moving some details of COBALD to Appendix A, presented at the end in the revised manuscript. For better understanding, we have divided Section 2.2.2 into two subsections (2.2.2.1 and 2.2.2.2) to separate the description of SOLAIR Boulder Counter from the methods used for the estimation of cloud microphysical properties in the revised manuscript. However, we have retained the description of CALIOP, CATS and Himawari-8 observations as they were, because their description is short compared to COBALD and Solair Boulder Counter.

Same comment for the description of ERA5, the justification to use this dataset could be shortened to lines 6,7, and added details such as number of levels available might be added information confusing for the reader (do we need these 37 levels here?)

Reply: There are different types of ERA5 data products available to the users. In the current manuscript, we have used ERA5 data products available at 37 pressure levels which is different

from ERA5 data products available at 137 model levels that provide better resolution in the TTL as compared to those available at few pressure levels. We have deleted this information on 37 levels in the revised manuscript.

Overall, this all method section from p4 to p12 would gain to be summed up to improve the understanding of the discussion regarding each result. Maybe a table, with each key variable and the corresponding instrument? This is just a suggestion.

Reply: We highly appreciate referee's idea of summarizing method section in the form of a table. We have included a table in the revised manuscript.

The lidar ratio part (starting p.17) first describes a discussion about different methods to retrieve this variable, when the key message of the paragraph, which is the authors methods and their demonstration of the use of COBALD and Boulder Counter to retrieve it, appears at the end. This paragraph might gain to be restructured/ shortened to guide the reader to its key points.

Reply: We have moved the discussion on different methods to retrieve lidar ratio to the Appendix in the revised manuscript.

p30, the paragraph from l12 to line 19 : As it is, it might be added to the introduction. It doesn't comment on the results, but introduces knowledge about literature regarding gravity waves, not directly connected to the author's findings that are described afterwards. Maybe rephrase/shorten it ? The introduction on the other hand is missing some references to existing literature about gravity wave impacts on cirrus clouds, as the authors are demonstrating it in this study.

Reply: We thank the referee for this suggestion. We have moved the paragraph from line 12 to line 19 of p30 to the introduction.

- 2) The authors present different formulas to calculate the ice water content and efficient coefficient, describe the differences/ similarities then recap all their results in a table page 19. Is there a conclusion regarding the differences between the effective diameters ?

Reply: We thank the referee for pointing this out. We have added a conclusion regarding the differences between the effective diameters.

- 3) figure 2 doesn't fit exactly the description in the text :

CL4 is said to have crystals smaller than 40 microns when I don't see the 20 or 25 microns curve (so the crystals are in fact smaller than 20 microns?). Same comment for CL5.

Reply: We thank the referee for pointing this out. This discrepancy is caused because Fig. 2c shows profiles of particle concentration at six different size channels expressed in radius (please see the figure caption) while the text describes the particle size in diameter. We have revised Fig. 2c and expressed particle size in diameter to be consistent through the manuscript.

- 4) figure 12 : the symbols for COBALD, CALIOP, CATS are almost indistinguishable on the figure.

Reply: We have increased the size of the symbols and changed their colours in the revised Fig. 12 to make them distinguishable.

- 5) figure 13 and comments from line 14-28 p.31 : The test of MERRA-2 accuracy could be just mentioned, more details would belong to an appendix. I am not sure what figure 13 is adding to the work already presented in the previous parts ? Maybe I am missing something here.

Reply: This figure is used to show the location of cold-point tropopause, unresolved by GEOS-5 temperature profiles, and the presence of tropopause cirrus clouds as observed by CATS, CALIOP, and COBALD shown in Figs. 11 and 12. It was difficult to locate the cold-point tropopause from Fig.12, so we have used near-co-located observations (within 400 km and 6 hours) with respect to CATS and CALIPSO profiles as shown in Figs. 11 and 12.  Each panel in Fig. 13 clearly shows the location of cold-point tropopause and presence of thin tropopause cirrus cloud layers.

- 6) Figure 14 is in section 3.4.3 which is highlighting the role of gravity waves in the formation of CL5. It describes that temperature anomalies are inferred by subtracting a monthly mean profile to the 23 August profile. I am not sure to understand why choosing a monthly time scale here when gravity waves usually have periods no longer than a few days ?

Reply: Thanks for this comment. The wind or temperature profile at any given instant is a combination of the background mean plus wave perturbation. To extract the wave perturbation, the background mean variation needs to be removed. There are various methods to achieve this, and it is quite common to remove the monthly mean, seasonal mean, or climatological mean profile to obtain the perturbation profile. Sometimes, individuals also apply a linear or polynomial fit to the instantly measured individual profile to remove the background. In our case, since we have sufficient data, we opted to remove the monthly mean profile. The perturbation profile, obtained after removing the mean, consists of waves with contributions from a broad spectrum, ranging from high frequency to internal period. There are several observational studies (e.g., Reinares Martínez et al., 2021; Kim et al., 2016; Kim and Alexander, 2015) which used monthly mean profile to estimate perturbation for studying the gravity waves.

- typos/formulations

p.13, l17 : extra space between 100 and unit

Reply: We have removed the extra space between 100 and unit in the revised manuscript.

p.13, l 19, p34 l22 : 'extremely cold temperature'. The 'extremely' is confusing, compared to what ?
Reply: We have deleted the words "an extremely cold" in the revised manuscript.

**References**

Kim, J.-E. and Alexander, M. J.: Direct impacts of waves on tropical cold point tropopause temperature, Geophysical Research Letters, 42, 1584–1592, https://doi.org/10.1002/2014GL062737, 2015.

Kim, J.-E., Alexander, M. J., Bui, T. P., Dean-Day, J. M., Lawson, R. P., Woods, S., Hlavka, D., Pfister, L., and Jensen, E. J.: Ubiquitous influence of waves on tropical high cirrus clouds, Geophysical Research Letters, 43, 5895–5901, https://doi.org/10.1002/2016GL069293, 2016.

Reinares Martínez, I., Evan, S., Wienhold, F. G., Brioude, J., Jensen, E. J., Thornberry, T. D., Héron, D., Verreyken, B., Körner, S., Vömel, H., Metzger, J.-M., and Posny, F.: Unprecedented Observations of a Nascent In Situ Cirrus in the Tropical Tropopause Layer, Geophysical Research Letters, 48, e2020GL090936, https://doi.org/10.1029/2020GL090936, 2021.

---

## Author Comment (AC3)

**Response to Referee's (Referee #3) Comments**

Review of "Investigating the role of typhoon-induced gravity waves and stratospheric hydration in the formation of tropopause cirrus clouds observed during the 2017 Asian monsoon" by Pandit et al.

This study analyzes backscatter sonde and optical particle counter measurements from balloons over Hyderabad, India to characterize properties of subvisible cirrus layer observed near the tropopause. They then investigate the formation mechanism of these cirrus cloud layers near the tropopause using a combination of back trajectories, satellite observations, and ERA5 reanalysis data. They conclude that overshooting convection from a typhoon created a hydration patch which was transported towards Hyderabad and ascended (via gravity waves) to form the cirrus layer. While the analyses of balloon data to characterize the cirrus cloud properties were interesting, the discussion on the formation mechanism of those clouds, particularly its ties to gravity waves, needs further development and clarification. Detailed comments are provided below.

Reply: First of all, we would like to thank the referee for carefully going through the manuscript and providing his/her constructive comments. In the following, our point-by-point response to the referee's comments (in black text) are presented in blue text.

Main Comments:

Although the combined use of various observations is generally a good idea, the frequent comparisons between measurements from different data sources at different locations and times (e.g., Table 2, Fig. 7, 10, 13) were difficult to interpret. This is especially problematic for this study since the phenomena of interest (i.e., overshooting convection, gravity waves) are small scale features that require coincident measurements. I suggest refining and limiting the selection of observations to only those that meet rigorous time and space matching criteria. I would also present the comparisons in a manner that is easily interpretable by the reader.

Reply: We thank the referee for bringing up this aspect. Tropopause cirrus clouds can be best studied with in-situ data (using balloons and aircraft) and ground-based and space-borne lidars owing to their high-sensitivity and high-vertical resolution. However, these observations are not available all the time and are limited to small spatial scales. With these limitations, we have tried our best to connect these small-scale observations with satellite and reanalysis data with the best possible coincidence available to us. It's challenging to get coincident measurements all the time at the desired location. This study investigates the role of a synoptic scale phenomenon (a typhoon) on the formation of tropopause cirrus clouds via typhoon-induced stratospheric hydration and gravity waves. It is to note that the impacts of overshooting convection and gravity waves associated with a typhoon are more intense, longer and have larger spatial extent in contrast to those of a mesoscale convective system. Several studies (as mentioned in the text: p30, lines 16-20) have suggested that semi-circular gravity waves generated from typhoon centre propagate horizontally through the troposphere, stratosphere, and the mesosphere in expanding spirals with horizontal wavelengths typically in the range of 50-500 km and periods from 1 hour to 1.6 days.

Our measurement coincidence criterion falls within this range. We further provide more details to address this aspect while responding to the referee's detailed comments.

I also found the discussion of the trajectory results to be somewhat incoherent. If the main purpose of using the backward trajectories is to show the influence from deep convection on cirrus clouds measured by the balloon, the most important quantity to show is the temperature and relative humidity evolution from the convection encounter to the measurement location. Trajectory results should be more concisely presented instead of across multiple figures. I would similarly reorganize other parts of the paper and make the paper more concise overall.

Reply: We thank the referee for raising this concern. We completely agree with the referee that the main purpose of using the back trajectories is to show the influence from deep convection on cirrus clouds measured by the balloon. In Fig. 5 and Fig. 6, we have shown this aspect. We also understand that the most important quantity to show is the temperature and relative humidity evolution from the convection encounter to the measurement site. We did show temperature along the trajectories in Figs. 11 and 12. Unfortunately, the relative humidity (RH) derived along the back-trajectories from the GEOS-FP data were not reliable in the UTLS region and therefore, we have not shown that in the manuscript. However, we have presented RH from Aura-MLS observations co-located with cirrus clouds observations from CALIOP at two locations intersecting the back-trajectories (see Fig. S10). In order to show the transport of water vapor from the typhoon towards Hyderabad, we have presented a Hovmöller diagram for water vapor mixing ratio from ERA5 reanalysis data (Fig. 10a).

Since we use 3D back-trajectories, we have shown the movement of air-parcels both horizontally (Fig. 5a and 11) and vertically (Figs. 5b, 12a and 12b) as a function of time and space. In addition to this, meteorological parameters such as temperature, vertical wind, and anvil clouds are also derived along the back-trajectories and shown in both horizontal and vertical directions. It is challenging to show these many parameters in just one or two figures. In each section, different aspects of the trajectories have been presented in different figures to support the findings of the investigation. We further provide more details about this aspect while responding to the referee's detailed comments.

Detailed Comments:

- P4: The temperature impact by tropical storms in Biondi et al. studies is fairly localized, and as such, I would not describe it as "a change in large-scale temperature fields in the TTL"

  Reply: We have removed that sentence in the revised manuscript.

- P11: Have the authors looked in the effects of trajectory uncertainties on the results? I'm assuming these are kinematic trajectories, whose dispersive nature may be concerning especially when calculating forward trajectories from UTLS levels.

  Reply: The dispersive nature of the trajectories is taken care of by initializing clusters of air-parcels which allow dispersion due to horizontal and vertical wind shears, representing

the synoptic-scale dynamics (Fairlie et al., 2014; Pierce and Fairlie, 1993). The back-trajectories from the Langley trajectory model (LaTM) have been extensively used to study the dispersion of volcanic plume (Fairlie et al., 2014) in the UTLS region and also to map UTLS aerosols to deep convective clouds observed from geostationary satellites (Fairlie et al., 2014; Vernier et al., 2015, 2018, 2021).

This study neither uses any LaTM forward trajectories nor represents any sub-grid scale convective or turbulent dispersion. Moreover, the back-trajectories used in this study to investigate the convective influence do not go back in time for more than 3 days. Thus, the uncertainties due to the dispersion would be reduced.

- P12: Is the use of ERA5 cloud cover fraction necessary? Cloud fields in reanalyses are model products (Wright et al., 2020) and vary depending on prescribed boundary conditions, physical parameterizations (e.g., convection), data assimilation approach, and assimilated data. It is likely unsuited as "cloud observations". Rather, plotting the 100 hPa temperature (or cold point temperature) field may be more reliable.

  Reply: We thank the referee for this suggestion. In the revised manuscript, we have removed cloud fraction from Section 2.4.3 and used 100 hPa temperature instead of cloud fraction in Figure 3 (a) as suggested.

- P15: What do you mean by running HYSPLIT trajectories from different altitudes of the CL5? Are the trajectories run at multiple altitudes within the ~2 km thick cloud layer estimated from the balloon measurements? And these trajectories intersect the CATS orbit several hours later at the same altitude as the cirrus detected from CATS? Can you be more specific?

  Reply: We calculated HYSPLIT forward trajectories at three different altitudes (16.8 km, 17.8 km, and 18.3 km) within the vertical extent of CL5. These altitudes correspond to the locations of CL5 near its base altitude (16.8 km), the cold-point tropopause (17.8 km) and near the top altitude (18.3 km) of CL5, respectively. These trajectories are intersected by the CATS orbit after ~2.5 hours of the passage of the air-parcels coming from the CL5. This means that the tropopause cirrus clouds were detected by the CATS after the passage of air-parcels coming from the CL5, indicating their possible influence. We have added this information in the main text.

- Fig. 5a: Can you explain what is meant by "day fraction (UTC)"? This is confusing to me since the dots are supposedly showing backward trajectories from 23 Aug 2017 at 20 UTC. Time since balloon measurement may be easier to interpret.

  Reply: We thank the referee for pointing this out. Each dot represents the location of an air-parcel in longitude-latitude space with colour showing its time in days. We have changed the colourbar title to "Days in Aug 2017."

- Fig. 5: What is the temperature history of these trajectories? Are the parcels cooling as they reach the balloon measurement point when cirrus clouds were formed? Figs. 11 and

12 are discussed much later, but this is a key parameter to look at as is RH with respect to ice.

Reply: The temperature history of these trajectories is discussed in detail in Section 3.4.3 after investigating the possible role of deep convective clouds and stratospheric hydration in influencing the air-parcels. As we can see in Figs. 11 and 12 that the air-parcels have experienced several cooling and warming phases near the tropopause before reaching the balloon site, the air-parcels have undergone cooling as they reach the balloon measurement point where tropopause cirrus clouds were detected.

We thank the referee for bringing up the discussion of relative humidity with respect to ice (RHi) here. The water vapour information along the back-trajectories from GEOS-FP data was not reliable near the tropopause which led to poor estimates of RHi along the back-trajectories and hence not shown here. However, the profiles of RHi observed from the Aura-MLS on 22 and 23 August 2017 co-located with CALIOP which intersected the trajectories show RHi values greater than 100 % at 100 hPa (~16.8 km) level, indicating supersaturation (see Fig. S10).

Fig. 5: How are the parcels that descend backward in time between 115-125 degE in panel (a) represented in panel (b)? In the plan view, it looks like they are wrapped in the typhoon at 125-130 degE, which does not seem to be consistent with panel (b)?

Reply: The trajectories originating between 16 and19 km as shown in Fig. 5(b) do appear to be wrapped in the typhoon at 125-130° E when we change the scale of y-axis, thus, it is consistent with Fig. 5a (see the figure below). Since TTL region is the focus of this manuscript, we have only shown altitude region between 15 km and 20 km in Fig. 5b.

[Figure]

Figure 1) Five-day back-trajectories initialized on 23 August 2017 at 20:00 UTC from the balloon measurement locations at different altitudes between 16 and 19 km represented by colored dots as a function of latitude and longitude with colour showing the days in August 2017. The locations of deep convective anvil tops observed after 21 August 2017 from the Himawari-8 brightness temperature images at 10.4 μm wavelength channel that intersected the back-trajectories are shown by black dots. Typhoon Hato track is shown by black outlined coloured circles connected by black line with colour showing the day fraction. The night-time CALIPSO orbit track on 20 August 2017 between 17:27 and 17:40 UTC is shown by cyan line with the location of the overshooting cloud top shown by red asterisk. (b) Five day back-trajectories represented as a function of longitude and altitude with colour showing their origin altitude. The locations of the anvil top altitude that intersected with the back-trajectories are represented by black dots. The locations of CALIPSO overpass and the overshooting cloud top altitude are shown by blue vertical line and blue asterisk, respectively.

- P21: How exactly is the convective anvil top altitude determined from the brightness temperatures to quantify the convective influence? The text mentions that this is discussed in Section 2.4.1, but I don't see any description of this methodology.

Reply: The uncertainty in the estimation of anvil cloud top altitude is about 0.75 km relative to CALIOP. Griffin et al. (2016) found that ~75% (65%) of MODIS (geostationary) overshooting top heights were within ±500 m of the coincident Cloud Profiling Radar (CPR/CloudSat)-estimated heights. Nearly all were within 1 km of CloudSat.

We thank the referee for pointing out this mistake. We have discussed the convective anvil top altitude estimation from the brightness temperature in Sect. 2.3.3. We have made this correction in the revised manuscript.

- P21: Pixel resolution certainly impacts temperatures within a convective cloud, but that fact alone is insufficient to claim that "the highest overshooting tops likely exceeded 18 km height". More concrete evidence of this is needed to make such a claim.

Reply: We understand the referee's concern here and this is why we used the word "likely" in the text. We tried to bring more confidence to Himawari-8 observations of cloud-tops by looking at nearest ground-based Doppler Weather Radar (DWR) observations available from IMD Machilipatnam station which also indicated towards the presence of such deep convective clouds. In lines 11-13 of Page 21, we have mentioned this. However, the radar images prior to 1200 UTC, when those convective clouds were at their peak altitude (as observed in Himawari-8 images), were unavailable to confirm the presence of overshooting tops exceeding 18 km. Moreover, the spatial scale and lifetime of those convective clouds were also small. In line 14-15, we also mention that we cannot fully rule out the probable influence of local convection.

- Table 2: While I understand the attempt to describe the temperature evolution of the parcel sampled, the disparate data sources (radiosonde, GNSS-RO), locations and times of these measurements make it difficult to say anything about the actual temperature evolution/effect, especially since we expect temperature perturbations on small spatial and temporal scales that greatly affect the cirrus cloud formation. It would make more sense to look at the temperatures along the backward trajectories to describe the evolution (like Fig. 11) and see if those agree with temperature observations at matching times and locations.

Reply: The main purpose of including radiosonde and GNSS-RO temperature profiles in Table 2 is to show the variation of cold-point tropopause altitude and temperature over Hyderabad region during the 24-hour period (23-24 August 2017) before, during, and after the balloon measurements. We do not have observations near the tropopause over Hyderabad all the time and this made us use the nearest possible (within 332 km) temperature profiles available from radiosonde and GNSS-RO. Except GNSS-RO observation, all radiosonde observations are within 136 km. Given the large horizontal extent (> 500 km) of the tropopause cirrus clouds as seen from CATS observations on 24 August 2017 at around 0136 UTC, we can say that the fluctuations in temperature may not be just limited to small spatial scales (see Fig. 1b for reference).

Regarding the referee's suggestion, the temperature used along the back-trajectories is extracted from GEOS-FP data which has limited vertical resolution to accurately infer the cold-point tropopause altitude. This is evident from Figure 13 (b) to (d) where we discussed

the presence of tropopause cirrus clouds observed from COBALD, CALIOP, and CATS along the back-trajectories while identifying the CPT altitude using nearest possible radiosonde and GNSS-RO observations.

- Fig. S3: (a) It would be good to include latitude and longitudes. (b) How realistic is the ERA5 water vapor (compared to observations like MLS)? Are you claiming that the wet anomalies are from the typhoon injected water and ice? (c) is x-axis the latitude?

  Reply: We have added latitude and longitudes in Fig. S3(a).

  The comparison of ERA5 water vapor with balloon-borne cryogenic frost-point hygrometer (CFH) during the Asian Summer Monsoon (ASM) has been discussed in Section 2.4.3. On an average, ERA5 water vapor overestimates by 0.7-0.9 ppmv compared to CFH (Brunamonti et al., 2019) and by ~ 2 ppmv compared to MLS observations (Khordakova et al., 2021; Sivan et al., 2022).

  The encircled wet anomalies from ERA5 at 1200 UTC shown in Fig. S3(b) are likely from the encircled deep convective clouds shown in Fig. S3(a).

  No, x-axis in Fig. S3(c) is longitude. We have labelled x-axis and y-axis in the revised figure.

- Fig. S4: Two temperature profiles are compared, but they are again from two different sources at two different locations at two different times. It is overreaching to claim from this figure alone that "tropopause temperature and tropopause altitude might be substantially reduced by the overshooting convection". Or is this meant to approximate the cold-point tropopause altitude?

  Reply: Yes, the two temperature profiles are just meant to approximate the cold-point tropopause altitude near the typhoon overshoot.

- Fig. 6: Do the back trajectory segments shown here exactly match the trajectories in Fig. 5b? It would be better if you could somehow combine these plots into one since they are approximately showing the same information.

  Reply: The back trajectory segments shown in Fig. 6 are the snapshot of back-trajectories on 22 August 2017 at 1900 UTC when they intersected CALIPSO overpass which observed a tropopause cirrus cloud layer in the outflow of typhoon *Hato*. These trajectories are different from those shown in Fig. 5b which showed 5-day back-trajectories initialized from the balloon measurement location between 16 and 19 km altitude. Adding this figure to Fig. 5 would make it difficult for the reader as there will be too much information in the same figure; we therefore decided to keep it separate.

- Fig. 7: Where are the MLS observations (track) on the map? It would be helpful to show those to interpret panel (b). Due to the deep averaging kernel of MLS, it is not clear that

the 6.5 ppmv water at 82.5 hPa actually represents enhanced water above the CPT as you suggest in p23. They could be contributed from enhanced moisture near the CPT.

Reply: We thank the referee for this suggestion. In the revised figure, we have added MLS track on the map.

We agree with the referee's concern on the deep averaging kernel of MLS and its impact on the magnitude of water vapor mixing ratio values near the CPT. We have rephrased this sentence in the revised manuscript.

- Fig. 9: Is (a) identical to Fig. S4? If so, this needs to be noted (and S4 could be removed). If not, please explain. How do the MLS values compare with the mean ERA5 water vapor shown in (b)? Rather than color coding the dots by the date, consider coloring the dots with the WV mixing ratios?

Reply: Yes, Fig. 9(a) is identical to Fig. S4 except the location of nearest radiosonde sounding, GNSS-RO sounding and the maximum anvil top height of typhoon Hato overshoot estimated from Himawari-8 images on 22 August 2017 at 0900 UTC shown on the map. We have removed Fig. S4 (top panel) in the revised supplementary material and added the location of GNSS-RO sounding and maximum anvil top height in Fig. 9(a) in the revised manuscript.

The ERA5 water vapor mixing ratio is overestimated by ~ 2 ppmv compared to MLS observation (Khordakova et al., 2021; Sivan et al., 2022).

Please note that the MLS tracks shown in Fig.9b are for different days between 20 and 23 August 2017 while the map shows two-day (22 and 23 August 2017) mean water vapor mixing ratio at 70 hPa pressure level from ERA5 showing the transport of enhanced water vapor. Hence, they are not compared. The magnitude of MLS water vapor mixing ratio at different pressure levels on different dates can be seen in Fig. 10b.

- Fig. 10: It would better to plot panels (b) to (e) in chronological order to see the decreasing magnitude (or select one representative location and combine the four profiles in one plot).

Reply: We thank the referee for this suggestion. We have modified the panels (b) to (e) in chronological order.

- P28: Have the authors checked to confirm that supersaturation (and how high of a supersaturation) is achieved along the trajectories to allow for ice nucleation?

Reply: Unfortunately, the relative humidity derived along the back-trajectories from the GEOS-FP data are not reliable in the UTLS region and therefore, we have not shown them in the manuscript. However, we have presented relative humidity with respect to ice (RHi) from Aura-MLS observations co-located with cirrus cloud observations from CALIOP at two locations intersecting the back-trajectories (see Fig. S10). Profiles of RHi observed from the Aura-MLS on 22 and 23 August 2017 co-located with CALIOP show RHi values

greater than 100 % at 100 hPa (~16.8 km) level, indicating supersaturation that resulted in ice formation. This is mentioned in the main text on p31, lines 25-27.

- Fig. 12: What resolution analyses are used for these trajectories? Is the resolution high enough to capture gravity waves? I see that the use of MERRA-2 reanalysis data is mentioned later on p31, but this should be discussed earlier along with Fig. 12. Also, how exactly are the vertical wind speeds derived along the trajectories in (b)? I presume the vertical winds are used to compute the kinematic trajectories.

    Reply: Three dimensional trajectories are computed using NASA Langley Trajectory Model (LaTM) which is driven by NASA Global Modeling and Assimilation Office (GMAO) Goddard Earth Observing System Version 5 (GEOS-FP) analyzed winds with a native horizontal resolution of 0.25° x 0.3125°. We use 6-hourly, time averaged wind fields resolved to horizontal resolution of 1.25° x 1° with 72 levels in the vertical having a resolution of ~1 km near the tropopause (Fairlie et al., 2014).

    MERRA-2 temperature data were not used in this study. We have corrected this error by changing "MERRA-2" to GEOS-FP in the revised manuscript.  We thank the referee for pointing it out.

    The vertical wind speeds are derived from latitude, longitude, altitude, and time information of the air-parcels. The vertical wind speed is used here to infer the upward or downward motion of the air-parcels which could indicate, respectively, towards cooling or warming associated with the air-parcels. Though the magnitude of the vertical wind speed may not be accurate in comparison to the observations (which we do not have), they do give a qualitative idea about the cooling and warming experienced by the air-parcels along the back-trajectories. It is interesting to see that the cooling and warming are associated with upward and downward motion of the air-parcels, respectively. Yes, vertical winds from the GEOS-FP are used to compute kinematic trajectories.

- Fig. 13: The purpose of this figure is unclear. The text seems to suggest that the purpose is to show the cold anomalies using high resolution data (and their impact cirrus), but it is difficult to interpret where and when these measurements are taken in relationship to the balloon measurements and trajectories.  The time and lat/lon are noted in the figure caption, that without a map or time series to place them in larger context, it is very difficult to interpret this figure.  It is even unclear whether the measurements plotted in one panel are coincident in time and location (they don't appear to be).

    Reply: This figure is used to show the location of cold-point tropopause, unresolved by GEOS-FP temperature profiles, and the presence of tropopause cirrus clouds as observed by CATS, CALIOP, and COBALD shown in Figs. 11 and 12. The purpose of Fig. 13 is clearly mentioned on page 32, lines 12-25. It was difficult to locate the cold-point tropopause from Fig. 12, so we have used nearest possible temperature observations (within 400 km and 6 hours) with respect to CATS and CALIPSO profiles as shown in Figs. 11 and 12.  This co-location criterion (within 400 km and 6 hours) is mentioned in

the main text (p31, line 18). Each panel in Fig. 13 shows clearly the location of cold-point tropopause and presence of thin tropopause cirrus cloud layers.

The location and date/time information of these measurements are shown on a map in Fig. 11. We provide the date/time and location of these measurements on each panel in the revised manuscript for clarity.

- Fig. 14: Temperatures were anomalously cold at ~17.9 km near the CPT on 23 Aug, but the tropopause altitudes presumably vary across all the profiles used to calculate the mean. Therefore, panel (a) alone is not sufficient to claim that the CPT was unusually cold on that day. It might make more sense to calculate the anomalies in tropopause relative coordinates. I also do not follow the argument that the variations above 15 km are likely due to waves ("wave-like pattern"). Cooling is likely associated with cirrus formation, but the suggested role of waves from Fig. 14 vertical profiles seems circumstantial at best. The authors mention a "Hodograph analysis of radiosonde data" to show evidence of gravity waves and derive their characteristics, but no description of the analysis is provided.

Reply: Thanks for raising this aspect. Please note that in Fig. 14, we are presenting perturbation profiles after removing the monthly mean profile, and there is a ~8°C cooling on August 23 (Fig. 14 a). While it is true that the CPT altitudes may not be the same on all the days which are used to estimate the mean profile, we only assert that there is significant cooling near the tropopause. This indicates that the background mean was much warmer on most days of that month. Since demonstrating that the CPT was the coldest is not the primary focus of this plot, we have not estimated the tropopause relative coordinates.

Regarding the wave-like pattern, please note that in all the perturbation profiles, a systematic variation with positive and negative values is evident, generally attributed to waves. The difference between two positive peaks (or negative peaks) gives the vertical wavelength of the wave. As there can be a superimposition of many waves, we observe smaller perturbations superimposed on larger perturbations. For more details on the extraction of wave properties and hodograph, please refer to Ratnam et al. (2006), specifically their Fig. 9. Additionally, for perturbation profiles in all three components (T, U, and V), refer to their Fig. 5.

The hodograph analysis is performed using a method described in Leena et al. (2012) which has already been mentioned in the text (page 35, lines 9-12). We have added the method description in sub-section 2.2.3 of the data and method section. We have also included a figure (Fig. S11) for the hodograph analysis in the revised supplementary information.

**References:**

Wright, J. S., Sun, X., Konopka, P., Krüger, K., Legras, B., Molod, A. M., Tegtmeier, S., Zhang, G. J., and Zhao, X.: Differences in tropical high clouds among reanalyses: origins and radiative impacts, Atmos. Chem. Phys., 20, 8989–9030, https://doi.org/10.5194/acp-20-8989-2020, 2020

Brunamonti, S., Füzér, L., Jorge, T., Poltera, Y., Oelsner, P., Meier, S., Dirksen, R., Naja, M., Fadnavis, S., Karmacharya, J., Wienhold, F. G., Luo, B. P., Wernli, H., and Peter, T.: Water Vapor in the Asian Summer Monsoon Anticyclone: Comparison of Balloon-Borne Measurements and ECMWF Data, Journal of Geophysical Research: Atmospheres, 124, 7053–7068, https://doi.org/10.1029/2018JD030000, 2019.

Fairlie, T. D., Vernier, J.-P., Natarajan, M., and Bedka, K. M.: Dispersion of the Nabro volcanic plume and its relation to the Asian summer monsoon, Atmospheric Chemistry and Physics, 14, 7045–7057, https://doi.org/10.5194/acp-14-7045-2014, 2014.

Griffin, S. M., Bedka, K. M., and Velden, C. S.: A Method for Calculating the Height of Overshooting Convective Cloud Tops Using Satellite-Based IR Imager and CloudSat Cloud Profiling Radar Observations, Journal of Applied Meteorology and Climatology, 55, 479–491, https://doi.org/10.1175/JAMC-D-15-0170.1, 2016.

Khordakova, D., Rolf, C., Grooß, J.-U., Müller, R., Konopka, P., Wieser, A., Krämer, M., and Riese, M.: A case study on the impact of severe convective storms on the water vapor mixing ratio in the lower mid-latitude stratosphere observed in 2019 over Europe, Dynamics/Field Measurements/Stratosphere/Physics (physical properties and processes), https://doi.org/10.5194/acp-2021-749, 2021.

Pierce, R. B. and Fairlie, T. D. A.: Chaotic advection in the stratosphere: Implications for the dispersal of chemically perturbed air from the polar vortex, Journal of Geophysical Research: Atmospheres, 98, 18589–18595, https://doi.org/10.1029/93JD01619, 1993.

Ratnam, M. V., Tsuda, T., Shibagaki, Y., Kozu, T., and Mori, S.: Gravity Wave Characteristics over the Equator Observed During the CPEA Campaign using Simultaneous Data from Multiple Stations, Journal of the Meteorological Society of Japan. Ser. II, 84A, 239–257, https://doi.org/10.2151/jmsj.84A.239, 2006.

Sivan, C., Kottayil, A., Legras, B., Bucci, S., Mohanakumar, K., and Satheesan, K.: Tracing the convective sources of air at tropical tropopause during the active and break phases of Indian summer monsoon, Clim Dyn, 59, 2717–2734, https://doi.org/10.1007/s00382-022-06238-9, 2022.

Vernier, H., Rastogi, N., Liu, H., Pandit, A. K., Bedka, K., Patel, A., Ratnam, M. V., Kumar, B. S., Zhang, B., Gadhavi, H., Wienhold, F. G., Berthet, G., and Vernier, J.-P.: Exploring the inorganic composition of the Asian Tropopause Aerosol Layer using medium-duration balloon flights, Atmospheric Chemistry and Physics Discussions, 1–30, https://doi.org/10.5194/acp-2021-910, 2021.

Vernier, J.-P., Fairlie, T. D., Natarajan, M., Wienhold, F. G., Bian, J., Martinsson, B. G., Crumeyrolle, S., Thomason, L. W., and Bedka, K. M.: Increase in upper tropospheric and lower stratospheric aerosol levels and its potential connection with Asian pollution, Journal of Geophysical Research: Atmospheres, 120, 1608–1619, https://doi.org/10.1002/2014JD022372, 2015.

Vernier, J.-P., Fairlie, T. D., Deshler, T., Ratnam, M. V., Gadhavi, H., Kumar, B. S., Natarajan, M., Pandit, A. K., Raj, S. T. A., Kumar, A. H., Jayaraman, A., Singh, A. K., Rastogi, N., Sinha, P. R.,

Kumar, S., Tiwari, S., Wegner, T., Baker, N., Vignelles, D., Stenchikov, G., Shevchenko, I., Smith, J., Bedka, K., Kesarkar, A., Singh, V., Bhate, J., Ravikiran, V., Rao, M. D., Ravindrababu, S., Patel, A., Vernier, H., Wienhold, F. G., Liu, H., Knepp, T. N., Thomason, L., Crawford, J., Ziemba, L., Moore, J., Crumeyrolle, S., Williamson, M., Berthet, G., Jégou, F., and Renard, J.-B.: BATAL: The Balloon Measurement Campaigns of the Asian Tropopause Aerosol Layer, Bulletin of the American Meteorological Society, 99, 955–973, https://doi.org/10.1175/BAMS-D-17-0014.1, 2018.

---

## Author Response (AR2)

**Author's Response to Editor's Decision**

We thank the editor for handling our manuscript and thank the referees for going through it carefully and providing their feedback. We have revised the manuscript based on the comments provided by referee #2 in his/her report #1. Point-by-point replies to the referee's comments are provided below. Comments are in black text while their replies are in blue text. We hope the revised version of the manuscript is suitable to be accepted for publication in Atmospheric Chemistry and Physics journal.

**Point-by-point reply to the referees' comments**

**Report#1 comments**

I thank the authors for their corrections and answers after the first round of review. I do think this paper has improved a lot and provides a nice observational study, unique in the wide set of techniques used.

Reply: We thank the referee for carefully going through our manuscript and providing his/her comments and suggestions for further improvement. We are glad that the referee has found many improvements in the manuscript and feels that it provides a nice and unique observational study.

I would still recommend a minor revision as I am still puzzled by the term 'gravity waves' used in their study. In my understanding, the authors are removing a background defined as a monthly mean to retrieve the 'gravity waves' perturbations (example figure 14). These waves are first described by the 'wave-like patterns' they produce, until the hodograph analysis, described at the end of the paper, that gives some of their characteristics. The authors are using the study of Kim and Alexander 2016 to justify their method and compare their results. In my understanding, Kim and Alexander removed a monthly mean to obtain the perturbations yes, but they are very specific about this technique only filtering the signals with a period less than 30 days, so gravity waves but also Kelvin waves and mixed Rossby-gravity waves as well. They do not link cold phases of gravity waves with cirrus clouds (unlike wrote p44 l8), but between cold phase of tropical waves and cirrus clouds, without distinguishing the type of waves. I would not expect gravity waves to have a period longer than a few days at these latitudes, and still do not understand completely why the authors used a monthly background if the goal was to highlight the gravity waves specifically. Although, the horizontal wavelength derived by the hodograph is in the upper end of the distributions of the observed gravity waves (Vincent and Alexander, 2000,https://doi.org/10.1029/2000JD900196). So I wonder if I missed something and the authors have enough elements to call these perturbations 'gravity waves' or if using a more generic term of waves would be better ? Maybe deriving the characteristics of the waves earlier would be better to name these waves for specifically 'gravity waves' in the paper.

Reply: We thank the referee for bringing this point to our notice. We have replaced the word "gravity waves" with "waves" in the revised manuscript.

---

## Author Response (AR3)

**Author's response to editor's decision**

**Editor's decision:**

Dear Authors,

Thank you for addressing the last concern of the reviewer. Given the referee's assessment and your response, I am pleased to accept the paper for publication, congratulations!

I have nevertheless noticed at least one remaining typo and suggest a reformulation. You also have the possibility to check the manuscript for typos and text improvements before it is transferred to copy-editing.

Sincerely,

Aurelien Podglajen p 26 l 15: it can be questioned that the ice crystals -> it can be questioned whether the ice crystals p 36 l 34: Himawai-8 -> Himawari-8

Reply: We sincerely thank the editor for accepting our manuscript for publication. We also thank the referees for providing their constructive assessment which tremendously helped in improving the manuscript.

We thank the editor for pointing out two above-mentioned typos which we have corrected in the revised manuscript.

As suggested by the editor, we have also looked for typos and text improvements in the manuscript as well as in the supplementary information. The revised manuscript and the supplementary information with track changes are merged herewith.

[revised manuscript text omitted]

Figure S5: (a) 10.4 μm infrared cloud-top temperature from Himawari-8 at 17:40 UTC on 20 August 2017 showing the overshoots during the development phase of typhoon *Hato*. Cyan line represents the CALIPSO orbit track that observed the overshoot on 20 August 2017 between 17:27 and 17:40 UTC. (b) CALIOP total attenuated backscatter coefficient at 532 nm browse image showing the location of the overshooting cloud over a region near 19.9° N and 124° E with its top reaching ~18.6 km altitude (Source: CALIPSO/NASA). Complete attenuation of backscatter signal below 14 km altitude followed by low value of brightness temperature at 12.05 μm from Imaging Infrared Radiometer (IIR) instrument onboard CALIPSO indicate the presence of deep convective clouds.

[Figure]

[Figure]

-Figure S6: (a) CALIOP depolarization ratio and (b) attenuated colour ratio along its orbit track on 22 August 2017 between 19:06 and 19:20 UTC (Source: CALIPSO/NASA). The location of the tropopause cirrus cloud layer is encircled red.

[Figure]

- Clouds with base altitude >16km (Laminar cirrus). Trajectories are between 16 and 19 km

Days in August 2017

**Figure S7: Time evolution of back-trajectories with orbit tracks of CALIOP and CATS colour coded with respect to their overpass time. Black dots along the orbit tracks represent the locations of cirrus clouds with their base altitude above 16 km.**

[Figure]

**Figure S8: Temperature history of the air-parcels along the back-trajectories initialized from the balloon measurement sites between 16 and 19 km on 23 August 2017 as a function of longitude and potential temperature. Each filled coloured circle represents temperature value at every 100 m vertical resolution at each hour. Different magenta symbols represent the location of cirrus clouds near the tropopause obtained from CATS on 23 August 2017 at 01:00 UTC (plus), CALIOP on 22 August 2017 at 19:00 UTC (open circle) and on 23 August 2017 at 07:00 UTC (filled circle) and COBALD on 23 August 2017 at 21:00 UTC (asterisk).**

[Figure]

**Figure S9: Vertical profile of GPS ascent speed (black line) of the balloon obtained from the radiosonde on 23 August 2017.**

[Figure]

**Figure S10: Profiles of relative humidity with respect to ice (RHi) from Aura-MLS instrument along the orbits that intersected the back-trajectories between 16°-19° N on (a) 22nd August 2017 and on (b) 23rd August 2017.**

[Figure]

[Figure]

**Figure S11: Perturbations in zonal (u') and meridional (v') wind speed for (top) troposphere (8.6-10.1 km) region and (bottom) stratosphere (22-24.5 km) region. The red contour in each panel is the variation of v' as a function of u' while the purple ellipse is the fit to the contour.**

**Hodograph analysis description:** Perturbations (u' and v') in zonal and meridional wind profiles are estimated from their anomaly profiles by fitting a fourth order polynomial. Hodograph analysis is performed by fitting an ellipse to u' versus v' plot for both troposphere (from 8.6 km to 10.1 km) and stratosphere (from 22 km to 24.5 km) regions, as shown in Fig. S11 above. From the fit, various wave (GW) parameters are estimated which are listed below in Table S1.

**Table S1: List of wave parameters obtained from hodograph analysis.**

| GW parameter/Region | Period (hour) | Vertical wavelength (km) | Horizontal wavelength (km) | Horizontal propagation direction (degree) | Brunt-Väisälä frequency | Horizontal phase speed (m/s) | Vertical phase speed (m/s) |
|---|---|---|---|---|---|---|---|
| Troposphere (1-14 km) | 24 | 1.5 | 702 | 12 | 0.01 | 8.1 | 0.017 |
| Stratosphere (18-26 km) | 26 | 2.5 | 1770 | 57 | 0.02 | 18.8 | 0.026 |

It is to note that the horizontal wavelength is 1770 km in the stratosphere and the direction of propagation is from south-east to north-west confirming that these waves are generated from typhoon *Hato*.